# Disintegration promotes protospacer integration by the Cas1-Cas2 complex

**Chien-Hui Ma[1], Kamyab Javanmardi[1], Ilya J Finkelstein[1,2]\*, Makkuni Jayaram[1]\***

[1]Department of Molecular Biosciences and Institute of Cell and Molecular Biology, University of Texas at Austin, Austin, United States; [2]Center for Systems and Synthetic Biology, University of Texas at Austin, Austin, United States

**Abstract** 'Disintegration'—the reversal of transposon DNA integration at a target site—is regarded as an abortive off-pathway reaction. Here, we challenge this view with a biochemical investigation of the mechanism of protospacer insertion, which is mechanistically analogous to DNA transposition, by the *Streptococcus pyogenes* Cas1-Cas2 complex. In supercoiled target sites, the predominant outcome is the disintegration of one-ended insertions that fail to complete the second integration event. In linear target sites, one-ended insertions far outnumber complete protospacer insertions. The second insertion event is most often accompanied by the disintegration of the first, mediated either by the 3'-hydroxyl exposed during integration or by water. One-ended integration intermediates may mature into complete spacer insertions via DNA repair pathways that are also involved in transposon mobility. We propose that disintegration-promoted integration is functionally important in the adaptive phase of CRISPR-mediated bacterial immunity, and perhaps in other analogous transposition reactions.

**\*For correspondence:**
ilya@finkelsteinlab.org (IJF);
jayaram@austin.utexas.edu (MJ)

**Competing interests:** The authors declare that no competing interests exist.

## Introduction

CRISPR-based bacterial immunity uses information from a prior infection stored in the chromosome to target and destroy invading viruses or plasmids. The adaptive phase of this defense system involves the deposition of short pieces of DNA (protospacers) from foreign nucleic acids into the chromosome as spacers—the storage units of immunological memory (*Hille et al., 2018*; *Marraffini, 2015*). The CRISPR locus has a (leader-[repeat-spacer]ₙ-repeat) organization, and the long primary transcript derived from it is processed by Cas (CRISPR-associated) and/or host nucleases into mature CRISPR-RNAs (crRNAs) (*Barrangou and Marraffini, 2014*; *Barrangou, 2015*; *Wiedenheft et al., 2012*; *Wright et al., 2016*). The crRNAs, together with Cas effector protein(s), assemble the interference complex for recognition of invading nucleic acids through concerted protein-nucleic acid and base-pairing interactions followed by their cleavage and destruction.

CRISPR-Cas systems are classified into two broad categories (*Brouns et al., 2008*; *Gasiunas et al., 2012*). Class 1 systems employ a multi-protein-RNA interference complex for target recognition, termed Cascade. Class 2 systems utilize RNA-guided target recognition and cleavage by a single protein, that is, Cas9 or Cas12. These two classes are further divided into several types and subtypes (*Wright et al., 2016*; *Makarova et al., 2015*; *Shmakov et al., 2015*). Self-targeting is avoided via multiple mechanisms, including recognition of a protospacer adjacent motif (PAM) immediately bordering the crRNA-complementary DNA sequence (*Gasiunas et al., 2012*; *Jiang and Doudna, 2017*; *Jinek et al., 2012*; *Elmore et al., 2016*; *Pyenson and Marraffini, 2017*; *Smargon et al., 2017*; *Wang et al., 2019*).

Although CRISPR-Cas systems differ markedly in the execution of immune defense, the adaptation phase (invading DNA → pre-spacer → protospacer → integrated spacer) is well conserved but still incompletely understood (*Amitai and Sorek, 2016*; *Krupovic et al., 2017*). The conserved Cas1 and Cas2 proteins are essential (and sufficient in some systems) to carry out spacer acquisition in

vivo (*Nuñez et al., 2015a*; *Yosef et al., 2012*). In the *Escherichia coli* (Type I-E) and *Enterococcus faecalis* (Type II-A) Cas1-Cas2 crystal structures, a pair of Cas1 dimers flank a central Cas2 dimer on either side in a butterfly arrangement, providing a ridge that can accommodate the pre-spacer DNA (that includes the PAM sequence), or the protospacer DNA (that lacks a complete PAM), with splayed out single strand termini (*Nuñez et al., 2015b*; *Wang et al., 2015*; *Xiao et al., 2017*). The *E. faecalis* complex reveals a second ridge that can be occupied by the acceptor DNA, suggesting its integration competence (*Xiao et al., 2017*). Other Cas proteins, such as Cas4, Csn2, and Cascade/Cas9, may improve spacer acquisition efficiency, while also contributing to the mechanism for discrimination of non-self (pre-spacer) from self (integrated spacer) (*Heler et al., 2015*; *Wei et al., 2015a*). In addition, bacterial nucleoid associated proteins such as IHF (integration host factor) assist the integration reaction and enhance target specificity in the *E. coli* Type I-E system (*Nuñez et al., 2016*). Protospacer generation in the Type I and Type II systems involves PAM recognition within the pre-spacer, but also the exclusion of a complete PAM from the protospacer to be integrated (*Heler et al., 2015*; *Wei et al., 2015a*).

Protospacer integration is mechanistically analogous to DNA transposition, with terminal 3′-hydroxyls attacking the target phosphodiester bonds at the leader-repeat (L-R) junction on one strand and the repeat-spacer (R-S) junction on the other (*Nuñez et al., 2015a*; *Figure 1A*). These transesterification reactions are carried out by the Cas1 active sites of the Cas1-Cas2 complex. Repair of the intermediate containing the (protospacer)-(repeat) strand fusions completes the reaction with a duplication of the repeat sequence, one copy each on either side of the spacer insert (*Figure 1A*). The nucleophilic attacks by the 3′-hydroxyls are thought to occur in a stepwise fashion with bending of DNA to facilitate the reaction (*Xiao et al., 2017*).

Crystal structures of Cas1, Cas2 proteins and of Cas1-Cas2 complexes (*Nuñez et al., 2015b*; *Wang et al., 2015*; *Xiao et al., 2017*; *Ka et al., 2018*; *Ka et al., 2016*), as well as cryo-electron microscopy (cryo-EM) structures of Cas1-Cas2-Csn2 complexes (*Wilkinson et al., 2019*), are consistent with the transposition mechanism of protospacer integration. However, they also raise a number of intriguing questions. Are the adaptation steps—pre-spacer recognition, its processing into protospacer, and protospacer insertion into the CRISPR locus—carried out by the same complex, perhaps in coordination with bacterial helicase/nuclease proteins (*Amitai and Sorek, 2016*; *Dillingham and Kowalczykowski, 2008*; *Levy et al., 2015*; *Kim et al., 2020*)? Or, are processing and insertion performed by separate complexes in spatially and/or temporally distinct steps? Does in vitro integration of a pre-processed protospacer by Cas1-Cas2 fully recapitulate the features of the in vivo reaction?

Here, we probe the mechanism of *S. pyogenes* (Spy) Cas1-Cas2 catalyzed integration in vitro. We show that partial (one-ended) integration is the most common reaction product. Attempts by Cas1-Cas2 to complete protospacer insertion by the integration of the second strand are most often accompanied by the reversal of integration (disintegration) of the first strand. As a result, complete integration of a protospacer into the target site is exceedingly rare. Nevertheless, the incomplete intermediates formed by disintegration may be rescued via repair pathways known to operate during DNA transposition. We conclude that disintegration is an intermediate step in at least one mode of integration, which may be accomplished by more than one pathway.

## Results

### Protospacer insertion into a nominal CRISPR locus present in a supercoiled plasmid

The in vitro insertion of a protospacer into a target site by the Spy Cas1-Cas2 proteins (*Wright and Doudna, 2016*) is inefficient, suggesting that the system may not fully represent the in vivo reaction. It may be limited by missing accessory components or because it is uncoupled from preceding DNA processing steps. We reexamined this reaction using a negatively supercoiled plasmid harboring the target site as 'leader-repeat-spacer' (*Figure 1A,B*).

The prominent outcome was the conversion of the supercoiled plasmid into a relaxed distribution (*Figure 1C,D*). The formation of a spacer integrant joined to the leader-proximal repeat end (L-R) on one strand and the spacer-proximal repeat end on the other (R-S) was barely detectable. Only a fraction of the product migrated as a nicked circle expected for a complete protospacer integration event or semi-integration events at either the L-R or R-S junction (*Figure 1C,D*). We confirmed the

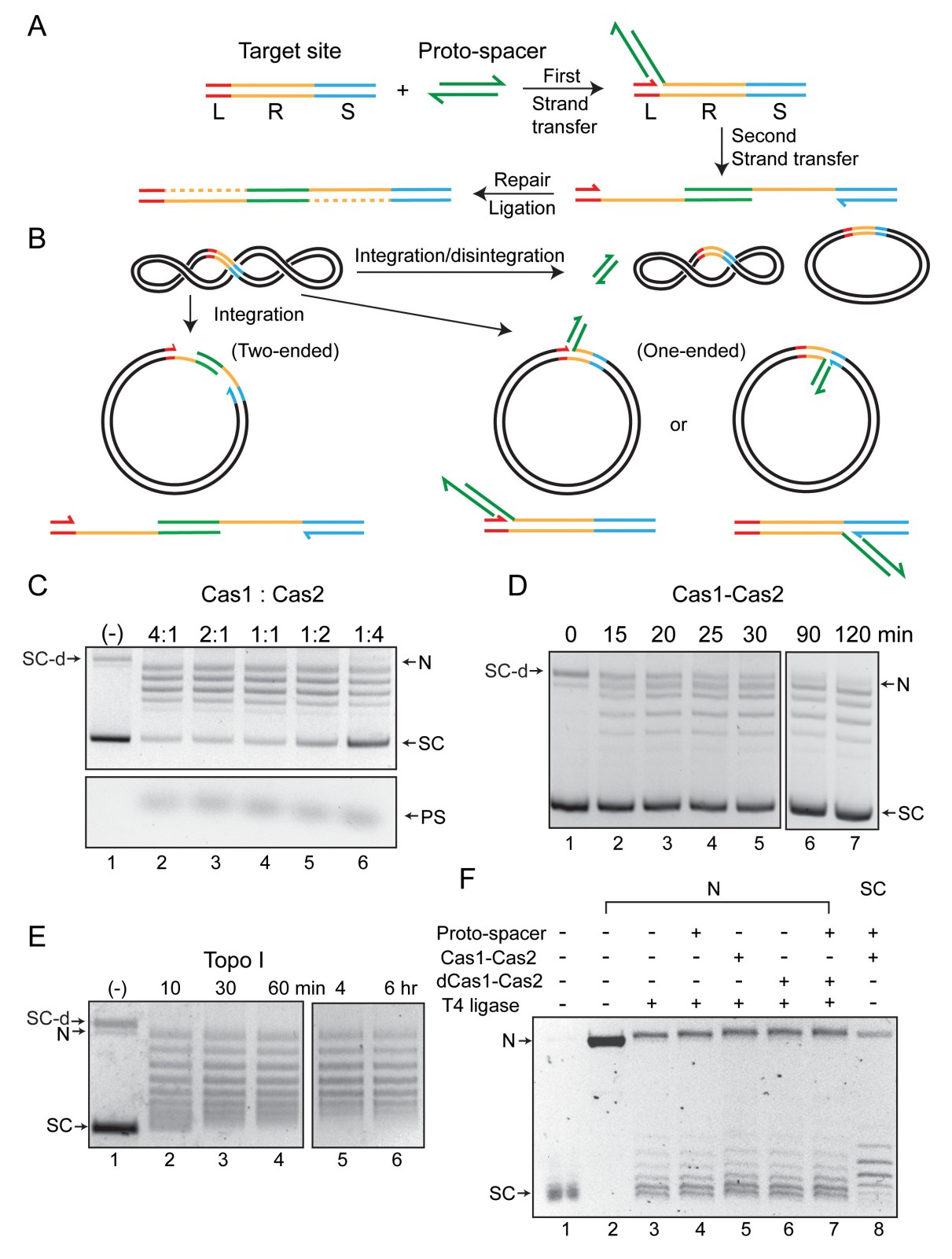

**Figure 1.** Cas1-Cas2 integrates and disintegrates protospacers at the CRISPR locus in a supercoiled plasmid. (A) Illustration of protospacer (PS, green) integration into a minimal CRISPR locus comprised of the leader (L, red), repeat (R, yellow), and spacer (S, blue). The reaction follows the cut-and-paste DNA transposition mechanism with a duplication of the repeat sequence. The 3'-hydroxyls that perform nucleophilic attacks at the L-R junction (top strand) and the R-S junction (bottom strand) are indicated by the split arrowheads. (B) In vitro reactions were performed with a supercoiled acceptor

*Figure 1 continued on next page*

*Figure 1 continued*

plasmid containing the minimal CRISPR locus (L=11 bp; R=36 bp; S=27 bp). Each strand of the protospacer was 26 nt long (DNA sequence in *Figure 2A*), with four single-stranded 3'-proximal nucleotides. The products of two-ended (complete) and one-ended (partial) protospacer integration and of protospacer integration-disintegration are diagrammed. Plasmid supercoiling will be reduced if the free DNA end undergoes rotation between integration and disintegration. (C) The target plasmid and the protospacer were reacted with Cas1-Cas2 mixtures in the indicated molar ratios. After 1 hr incubation, reactions were analyzed by agarose gel electrophoresis and ethidium bromide staining. A split view of the top and bottom portions of the gel is presented to show the plasmid and protospacer bands. (D) Reactions similar to those shown in C utilized a Cas1 to Cas2 molar ratio of 2:1 and were analyzed over a time course. (E) The substrate plasmid used for the reactions in (C) and (D) was treated with a sub-optimal amount of *Escherichia coli* topoisomerase I to follow the pattern of DNA relaxation over time. (F) The T4 ligase reactions (lanes 3–7) were performed on the plasmid containing the L-R-S target site nicked with Nb.BtsI (lane 2). The Cas1-Cas2 integration-disintegration reaction (lane 8) utilized the same plasmid in its supercoiled form (lane 1). Reactions were analyzed by gel electrophoresis in the presence of chloroquine (0.4 μg/ml). The DNA bands from the T4 ligase (lanes 3–7) and Cas1-Cas2 (lane 8) reactions are offset by one-half twist/writhe, as indicated by their staggered patterns. N = nicked plasmid; PS = protospacer; SC = supercoiled plasmid; SC-d = supercoiled plasmid dimer .

The online version of this article includes the following figure supplement(s) for figure 1:

**Figure supplement 1.** Integration-disintegration by *Streptococcus pyogenes* (Spy) Cas1-Cas2 is associated with plasmid relaxation.
**Figure supplement 2.** Cas1-Cas2 unwinds DNA within the target site during protospacer integration-disintegration.

presence of the protospacer in the 'nicked' plasmid band by labeling it with Cy5 (*Figure 1—figure supplement 1A*). By contrast, the Cy5-labeled protospacer was not inserted into the relaxed plasmid. Supercoil relaxation by Cas1-Cas2 required the CRISPR locus to be present in the target plasmid, and was dependent on the protospacer (*Figure 1—figure supplement 1B,C*). Optimal reaction occurred at 4:1 and 2:1 molar ratios of Cas1:Cas2, with increasing Cas2 inhibiting the reaction mildly (*Figure 1C*). These results are consistent with previous biochemical and structural data suggesting $(Cas1)_4$-$(Cas2)_2$ as the active insertion complex, with Cas1 performing the catalytic steps of protospacer strand transfer to the CRISPR locus (*Nuñez et al., 2015b*; *Wang et al., 2015*; *Figure 1—figure supplement 1B–D* and accompanying Supplementary Text S1).

The predominance of the relaxed (closed circular) plasmid over the 'nicked' plasmid in the product indicates that the Cas1-Cas2 complex fails to complete protospacer integration in most cases. The major in vitro activity of Cas1-Cas2 is the Sisyphean reversal of semi-integration events (disintegration) mediated by the 3'-hydroxyl exposed next to the integration junction.

## Topological features of the Cas1-Cas2 reaction in supercoiled plasmids

We employed topological analyses to address potential structural distortions of target DNA induced by Cas1-Cas2 during protospacer integration (*Nuñez et al., 2015b*; *Wang et al., 2015*; *Xiao et al., 2017*). We probed linking number changes (ΔLks) accompanying Cas1-Cas2-mediated integration-disintegration in a supercoiled target plasmid and T4 ligase-mediated strand closure in the nicked form of the same plasmid with or without bound Cas1-Cas2. Provided the Cas1-Cas2-induced topological stress within a target site is stable throughout the reaction, it will be preserved in the plasmid topoisomers formed by integration-disintegration (*Figure 1B,C*). In principle, the target topology in a Cas1-Cas2-bound nicked plasmid may also be captured by resealing the nick with DNA ligase.

DNA relaxation by Cas1-Cas2 occurred progressively, although not in a regular stepwise fashion (*Figure 1D*). The disappearance with time of the faster-migrating bands formed early in the reaction, and the corresponding increase in the more relaxed distribution, is consistent with more than one 'semi-integration-disintegration' event occurring in the same plasmid molecule. The relaxation pattern was distinct from a topoisomerase I-generated ladder (ΔLk=+1) (*Figure 1E*), suggesting that the broken strand goes through multiple rotations before it is resealed.

Integration-disintegration by Cas1-Cas2 and nick closure by T4 DNA ligase in the absence of Cas1-Cas2 resulted in similar, but not identical, plasmid topoisomer distributions (*Figure 1—figure supplement 1E*). The slight shift between the centers of the two distributions is consistent with supercoiling strain (writhe or twist) being sequestered by Cas1-Cas2 interaction with DNA. However, the possibility that relaxation by Cas1-Cas2 had not reached equilibrium cannot be ruled out. The relaxed plasmid topoisomers formed by ligation (without bound Cas1-Cas2) reflect differences in twist or writhe in individual DNA molecules due to thermal fluctuations. As the gel fractionation did not include an intercalating agent, the position of a band below the slowest migrating fully relaxed form (ΔLk=0) indicates only its absolute deviation from the relaxed state. A molecule with ΔLk=+1

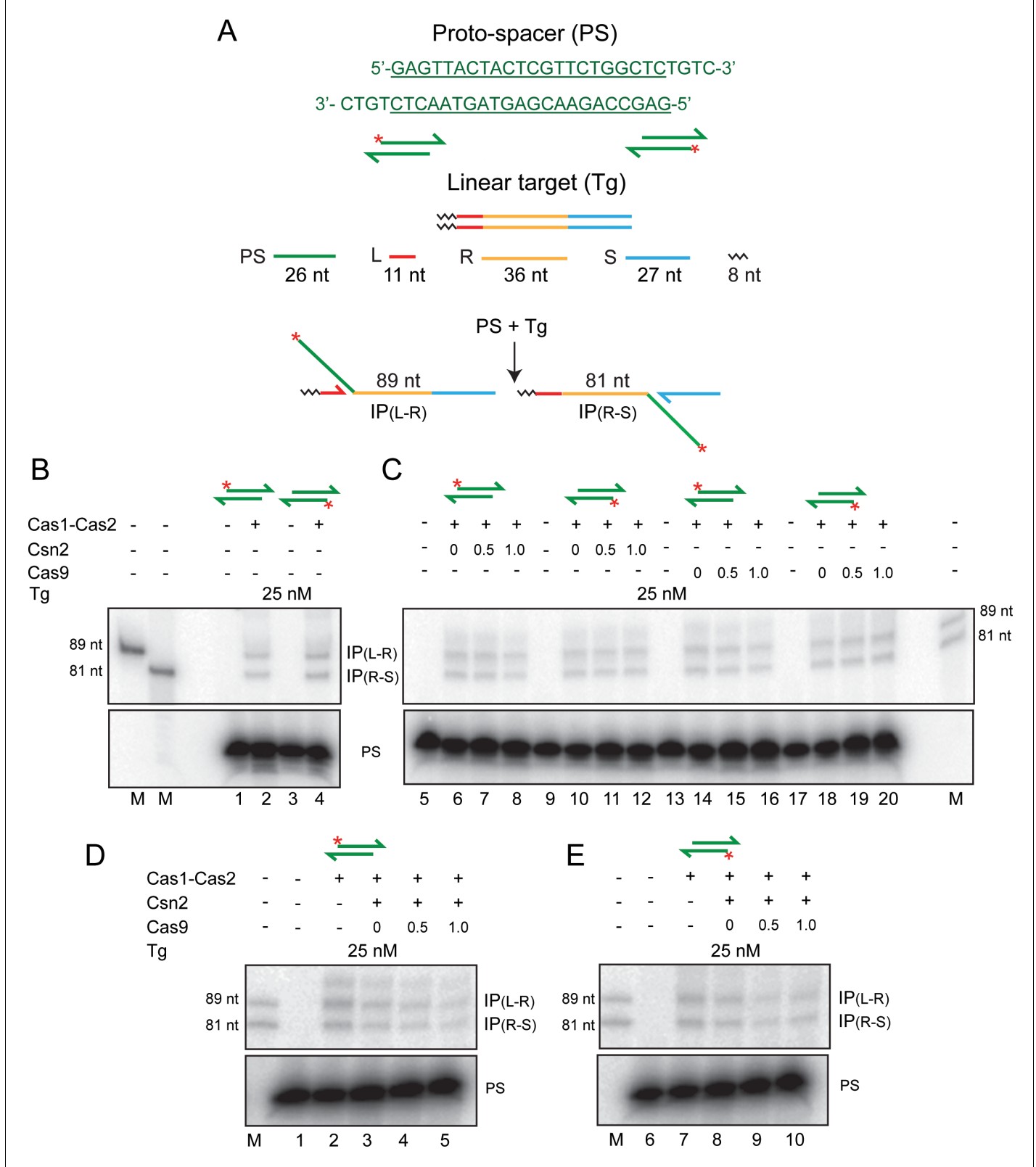

**Figure 2.** Cas1-Cas2 integrates a protospacer into a linear DNA target. (**A**) The protospacer sequence, with the double-stranded region underlined, is shown at the top. In the schematics, the protospacer ($^{32}$P-labeled at the 5'-end; asterisk) and the linear target site are color-coded as in **Figure 1**. The labeled products of integration at the L-R and R-S junctions are predicted to be 89 nt and 81 nt, respectively. (**B–E**) Reactions were performed using the indicated proteins, the protospacer labeled on the top or bottom strand and the unlabeled target. For all reactions, Cas1:Cas2 molar ratio was held at

*Figure 2 continued on next page*

*Figure 2 continued*

2:1. The numbers above individual lanes against Csn2 or Cas9 indicate their molar ratios to Cas1. In (**D** and **E**), the molar ratio of Csn2:Cas1 was 1:1. The reactions were analyzed by electrophoresis in 12% denaturing polyacrylamide gels, and bands were visualized by phosphorimaging. Only the portions of the gel containing the relevant products and the unreacted protospacer are shown here. The markers in lane M are synthetic oligonucleotides identical in sequence to the predicted integration products. IP = integration product; PS = protospacer.

The online version of this article includes the following figure supplement(s) for figure 2:

**Figure supplement 1.** Protospacer integration by Cas1-Cas2 into a supercoiled plasmid target is not stabilized by Csn2, Cas9, or both.

**Figure supplement 2.** A nicked linear target is active in protospacer integration by Cas1-Cas2.

**Figure supplement 3.** PAM-containing pre-spacer mimics do not confer strand selectivity on integration by Cas1-Cas2.

(positively supercoiled) and one with $\Delta Lk=-1$ (negatively supercoiled) would show the same migration.

Next, we analyzed the integration-disintegration and nick ligation reactions by separating DNA topoisomers in the presence of chloroquine. Unwinding of the double helix by the intercalator introduces compensatory positive supercoils in covalently closed circular molecules. Each topoisomer is separated from its nearest neighbors (migrating above and below) by one supercoil, the faster migrating one being more positively supercoiled. Strand sealing in the nicked plasmid was performed by T4 ligase without added factors (*Figure 1F*; lane 3) or after pre-incubation with the indicated Cas proteins and/or the protospacer (*Figure 1F*; lanes 4–7). Nearly identical topoisomer distributions were obtained after ligation in the absence of Cas1-Cas2 (*Figure 1F*; lane 3) or in the presence of either Cas1-Cas2 (*Figure 1F*; lane 5) or the catalytically inactive dCas1-Cas2 (*Figure 1F*; lane 6). Addition of protospacer by itself (*Figure 1F*; lane 4) or together with dCas1-Cas2 (*Figure 1F*; lane 7) to the ligase reaction did not alter this distribution. The topoisomer distribution of the Cas1-Cas2 plus protospacer treated supercoiled plasmid (*Figure 1F*; lane 8) was more negatively supercoiled ($\Delta Lk$ between $-1$ and $-2$) than that obtained from nick closure in the presence of dCas1-Cas2 and protospacer (*Figure 1F*; lane 7) (*Figure 1—figure supplement 2*).

The above results suggest that Cas1-Cas2 either stably constrains local negative supercoiling when bound to a target site in a supercoiled plasmid (without, however, impeding free rotation of the strand nicked by Cas1), or has had insufficient time to effect complete plasmid relaxation. For a target site present in a nicked plasmid, the Cas1-Cas2-induced topological stress is presumably dissipated via the nick before being sealed off by ligation. The observed $\Delta Lk$ between the Cas1-Cas2 relaxed and T4 ligase sealed plasmid distributions (*Figure 1F*; *Figure 1—figure supplement 2*) is consistent with the unwinding of a little more than one DNA turn within the target sequence during protospacer integration. This torsional strain may promote the formation of the repair-ready intermediate in which single-stranded repeat sequences flank the inserted protospacer (*Figure 1A*).

## Protospacer insertion into a linear target: Effects of Csn2 and Cas9 on integration/disintegration

Previous studies demonstrated the semi-integration of a protospacer into a linear DNA target, and indirectly suggested full integration (*Wright and Doudna, 2016*). The effect of DNA topology on the disintegration reaction could account for the apparent difference in the activities of supercoiled and linear targets in integration. In order to characterize the integration steps in further detail, we turned to reactions with linear substrates (*Figure 2A*).

In accordance with previous findings, we observed insertions of the 5′-end-labeled protospacer at the leader side (L-R junction) (89 nt band) or at the spacer side (R-S junction) (81 nt band) (*Figure 2B*). L-R and R-S insertions were approximately equally prevalent, although L-R insertions were more abundant at very short time points. This is consistent with semi-integration events initiated at L-R being converted to full-integration events by R-S insertion, as was suggested previously (*Wright and Doudna, 2016*). However, alternative explanations cannot be ruled out (see below).

Unlike orientation-specific protospacer integration in vivo, in vitro Cas1-Cas2 reactions showed no strand-specificity (*Figure 2B*). This bias-free insertion of the top or bottom strand from the protospacer was unchanged by the addition of Csn2 or Cas9 or both to the reactions (*Figure 2C–E*). These proteins, singly or in combination, also failed to stabilize protospacer integrations in the supercoiled plasmid target (*Figure 2—figure supplement 1*). Instead, they inhibited plasmid

relaxation. Inhibition could occur at the level of integration per se or strand rotation during integration-disintegration.

In order to address whether the products of integration could be stabilized by preventing potential disintegration, we used target sites with nicks placed within the leader and the spacer 3 nt away from the normal integration sites (*Figure 2—figure supplement 2*). The short 3 nt product (5′GAG-OH3′ or 5′GCA-OH3′) formed during integration would not be stably hydrogen-bonded to the complementary strand and is expected to diffuse away from the reaction center. The increased spacing of the 3′-hydroxyls of the shortened leader or spacer from the integration junction would block or curtail disintegration.

The nick-containing target yielded, in addition to the 81 nt and 89 nt products expected from the standard linear target (*Figure 2—figure supplement 2A,B*), product bands migrating slightly higher than each (*Figure 2—figure supplement 2A,C*). These extra bands, though uncharacterized, are consistent with potential integration events at the two phosphodiester positions between an authentic phosphodiester target and its proximal nick (Supplementary Text S2 in *Figure 2—figure supplement 2*). The flexibility afforded by the nick could presumably facilitate these aberrant integration events. The amount of the integration product (normal plus aberrant) was increased two- to three-fold in comparison to the normal product formed from the target without a nick. Although disintegration by the leader or spacer 3′-hydroxyl is reduced in the nicked target site, water-mediated disintegration could still be an impediment to stable integration (see below). The overall yield of disintegration could thus be limited.

The similar amounts of L-R and R-S insertion product bands formed in the linear target (*Figure 2B–E*) may arise from equal and independent one-ended insertions in the same target or in separate targets, or from completion of two-ended insertions within individual targets. A third possibility is that, within an intermediate containing a semi-integrant, integration of the second strand is coupled to the disintegration of the first. In this case, the one-ended integrant will oscillate between the L-R and R-S junctions. As a result, the net integration events at L-R and R-S will be equal at a steady state. Their absolute amounts would be dictated by how strongly they are destabilized by disintegration under a given set of reaction conditions. The difference between nearly zero and readily detectable integration in supercoiled versus linear targets can be accounted for by this interpretation.

## Cas1-Cas2 activity on pre-spacer mimics carrying the PAM sequence

The strand cleavage and strand transfer steps of protospacer insertion at the CRISPR locus must engender safeguards against self-targeting of the inserted spacer as well as its non-functional orientation. However, no strand selectivity is seen in the in vitro Cas1-Cas2 reactions with already processed protospacers (*Figure 2*; see also *Figure 3*). By coordinating PAM-specific cleavage of a pre-spacer with the transfer of this cleaved strand to the L-R junction, the inserted spacer will be in the correct orientation to generate a functional crRNA. To examine this possibility, we tested the integration characteristics of pre-spacer mimics containing the PAM sequence.

The inclusion of PAM or PAM and its complement in the integration substrates (*Figure 2—figure supplement 3A*) did not confer strand specificity on reactions with Cas1-Cas2 alone or with added Csn2, Cas9, or both (*Figure 2—figure supplement 3B–E*). Optimal integration by Cas1-Cas2 occurred with the 26 nt strands of the native protospacer with their 4 nt 3′-overhangs (*Figure 2—figure supplement 3B–E*; lanes 2). The pre-spacer mimics containing one or both >26 nt strands had reduced integration competence (*Figure 2—figure supplement 3B–E*; lane 4). Even here, the 26 nt strand with the 4 nt overhang (*Figure 2—figure supplement 3C*; lane 4) was preferred in integration over the longer 29 nt PAM-containing strand (*Figure 2—figure supplement 3D*; lane 4) or the 33 nt PAM complement-containing strand (*Figure 2—figure supplement 3E*; lane 4). In contrast to the processed protospacer that gave nearly equal integration at L-R and R-S, IP(L-R) ≈ IP(R-S) (*Figure 2—figure supplement 3B–E*; lane 2), the longer pre-spacer mimics were inhibited in integration at R-S, IP(L-R)>IP(R-S) (*Figure 2—figure supplement 3B–E*; lane 4). This is the expected outcome if the initial strand transfer occurs at L-R, and a ruler-like mechanism orients the reactive 3′-hydroxyl for the second strand transfer at R-S. This sequential two-step scheme for protospacer integration is consistent with the results shown in *Figure 3* as well. These reaction features of the pre-spacer mimics were not modulated by Csn2 or Cas9 (*Figure 2—figure supplement 3B–E*; lanes 6 and 8), although Csn2+Cas9 was inhibitory to integration (*Figure 2—figure supplement 3B–E*; lane 10).

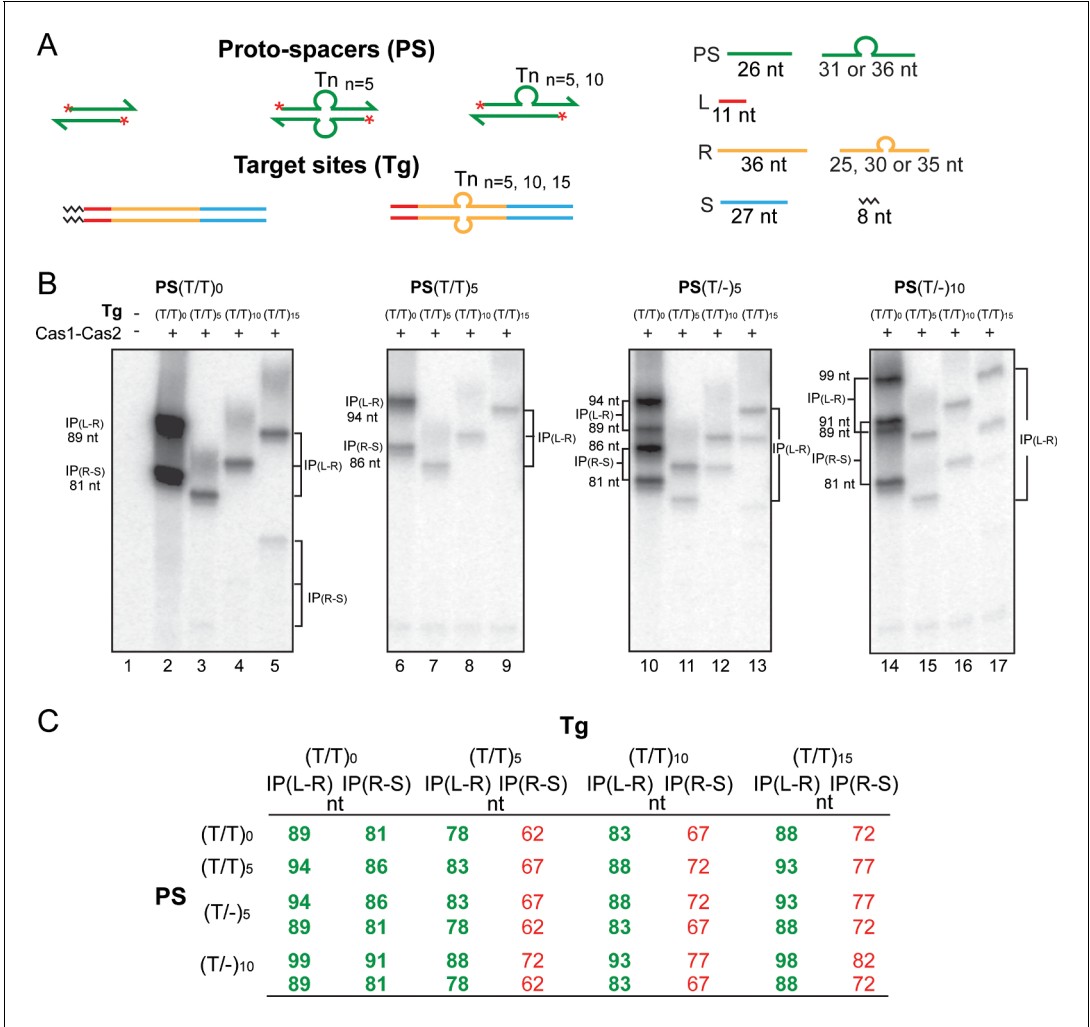

**Figure 3.** Cas1-Cas2 integrates modified protospacers into modified target sites. (**A**) Schematics of the native and modified versions of the protospacers (labeled at the 5′-ends on both strands; asterisks) and target sites used for integration assays are shown. (**B**) The reactions were performed and analyzed as described in *Figure 2* (see *Figure 3—figure supplement 1* for uncropped gel images). (**C**) The expected integration products from the various protospacer and target site pairings are summarized. The $(T/T)_0$ designation refers to the unmodified protospacer and target. The observed products from the reactions shown in (**B**) are in bold green font; those not detected or detected in extremely small amounts are in normal red font. IP = integration product.

The online version of this article includes the following figure supplement(s) for figure 3:

**Figure supplement 1.** Cas1-Cas2 integrates protospacers predominantly at the L-R junction in modified target sites.

There is no evidence for integration accompanying PAM-specific cleavage in our in vitro reactions. In the *E. coli* CRISPR system, Cas1-Cas2 is apparently sufficient for PAM-specific cleavage in vitro (*Wang et al., 2015*). By contrast, in the *S. pyogenes* system, cleavage is attributed to Cas9 or as yet uncharacterized bacterial nuclease(s) (*Jakhanwal et al., 2021*). The mechanism for generating an integration-proficient and orientation-specific protospacer, which may not be conserved among CRISPR systems, is poorly understood at this time (Supplementary Text S3 in *Figure 2—figure supplement 3*).

## Sequential and coordinated strand transfer during protospacer insertion by Cas1-Cas2

Current evidence suggests independent recognition of L-R and R-S junctions by Cas1-Cas2 but initiation of integration at L-R (*Wright and Doudna, 2016*). We examined the order of strand transfer between the two junctions using modified protospacers and target sites.

The modified protospacers contained base insertions either in both strands $(5'T3'/3'T5')_n$ or in only one strand $(5'T3'/-)_n$, with no additional changes from the native sequence (*Figure 3A*). As a result, the length of a modified strand was increased from the normal 26 nt by the number of inserted Ts. However, the 22 bp double-stranded region and the 4 nt 3'-overhangs at either end were retained as in the native protospacer. In the modified target sites, $(5'T3'/3'T5')_n$ insertions replacing a central 16 bp segment of the repeat were flanked on either side by 10 bp each of the leader-proximal and spacer-proximal repeat sequences. Consequently, the repeat lengths varied from 25 nt to 30 nt and 35 nt on each strand in the individual insertion derivatives (compared to 36 nt fully complementary strands in the native target). Based on the results from prior studies (*Wright and Doudna, 2016*; *Wei et al., 2015b*), the L-R and R-S junctions of the modified target sites were expected to be integration-competent. Furthermore, the premise that the inherent structural pliability of $(T/T)_n$ or $(T/-)_n$ would at least partly compensate for changes in the absolute lengths of the Cas1-Cas2 substrates was borne out by the experimental results (*Figure 3B,C*).

The native $(T/T)_0$-protospacer as well as the ones carrying T-insertions, $(T/T)_n$ or $(T/-)_n$, gave the predicted integration products at the L-R and R-S junctions with the native $(T/T)_0$-target site (*Figure 3B*; lanes 2, 6, 10, and 14). As the two strands of a $(T/-)_n$-protospacer were unequal in length, their integration at each junction resulted in two products with different gel mobilities (four bands in lanes 10 and 14). For the $(T/T)_5$, $(T/T)_{10}$, and $(T/T)_{15}$-targets, integration occurred almost exclusively at the L-R junction (*Figure 3B*; lanes 3–5, 7–9, 11–13, and 15–17). The corresponding products at the R-S junction were at or below the levels of detection (*Figure 3B*; *Figure 3—figure supplement 1*). Extended phosphorimaging brought to light weak bands shorter than the predicted authentic integration products (*Figure 3—figure supplement 1*). Although they are not directly relevant to our analysis, their possible origin is considered in Supplementary Text S4 in *Figure 3—figure supplement 1*.

Reactions of the modified protospacers with the modified target sites unveil a strict hierarchy in the choice of the L-R and R-S junctions during protospacer acquisition. *Figure 3C* differentiates the experimentally observed products from the undetected (or highly underrepresented) potential products. The strong L-R preference could occur at the level of target recognition (which then triggers catalysis) or at the level of catalysis (with no preferential recognition of either the L-R or the R-S junction). Structural snapshots of the *E. faecalis* Cas1-Cas2 integration complex suggests that, following a stochastic L-R/R-S search, preferential interaction with the L-R junction is established (*Wang et al., 2015*). Catalysis of the initial one-ended insertion at this target site follows. Within this intermediate, a ruler-like mechanism may be used to promote R-S junction interactions that culminate in the second strand transfer. When the search fails to identify an appropriately positioned R-S, as would be the case with the modified targets, the reaction is limited almost entirely to the L-R junction.

## Protospacer insertion into half-target sites

One question raised by the results from the modified target substrates is whether the (L-R)-before-(R-S) rule in protospacer strand transfer is manifested only when both these junction sequences are present in cis within a target site. Previous mutational analysis suggests that either the L-R or the R-S junction is used equally well for integration when only one of them is functional (*Nuñez et al., 2015b*; *Wang et al., 2015*). We now assayed the integration of a protospacer into linear L-R and R-S half-target sites to test R-S functionality in the absence or presence of L-R (*Figure 4* and *Figure 4—figure supplement 1*).

In reactions with the individual L-R and R-S half-sites, insertion products were obtained with either half-site alone (*Figure 4*; lanes 1–14). Consistent with the independent recognition of the L-R and R-S junctions by the Cas1-Cas2 complex, an R-S half-site was co-dominant with the 'minimal' L-R half-site (containing 18 bp of the repeat; *Figure 4—figure supplement 1*) when both were present in the same reaction (*Figure 4*; lanes 15–18). However, the extension of the repeat segment in the minimal L-R half-site by 4 bp, L-R(+4) (*Figure 4—figure supplement 1*), restored L-R dominance to the same degree as that displayed by the modified full linear targets (*Figure 4*; lanes 19–22). While the insertion product was formed from L-R(+4), none was detected from either the minimal R-S half-site (containing 18 bp of the repeat; *Figure 4—figure supplement 1*) or the R-S(+6) half-site (containing additional 6 bp of the repeat; *Figure 4—figure supplement 1*).

Thus, the hierarchical dominance of L-R seen in the full-target sites (*Figure 3B*) is lost when R-S is physically unlinked from the minimal L-R. However, L-R(+4) is dominant over R-S or R-S(+6) even in

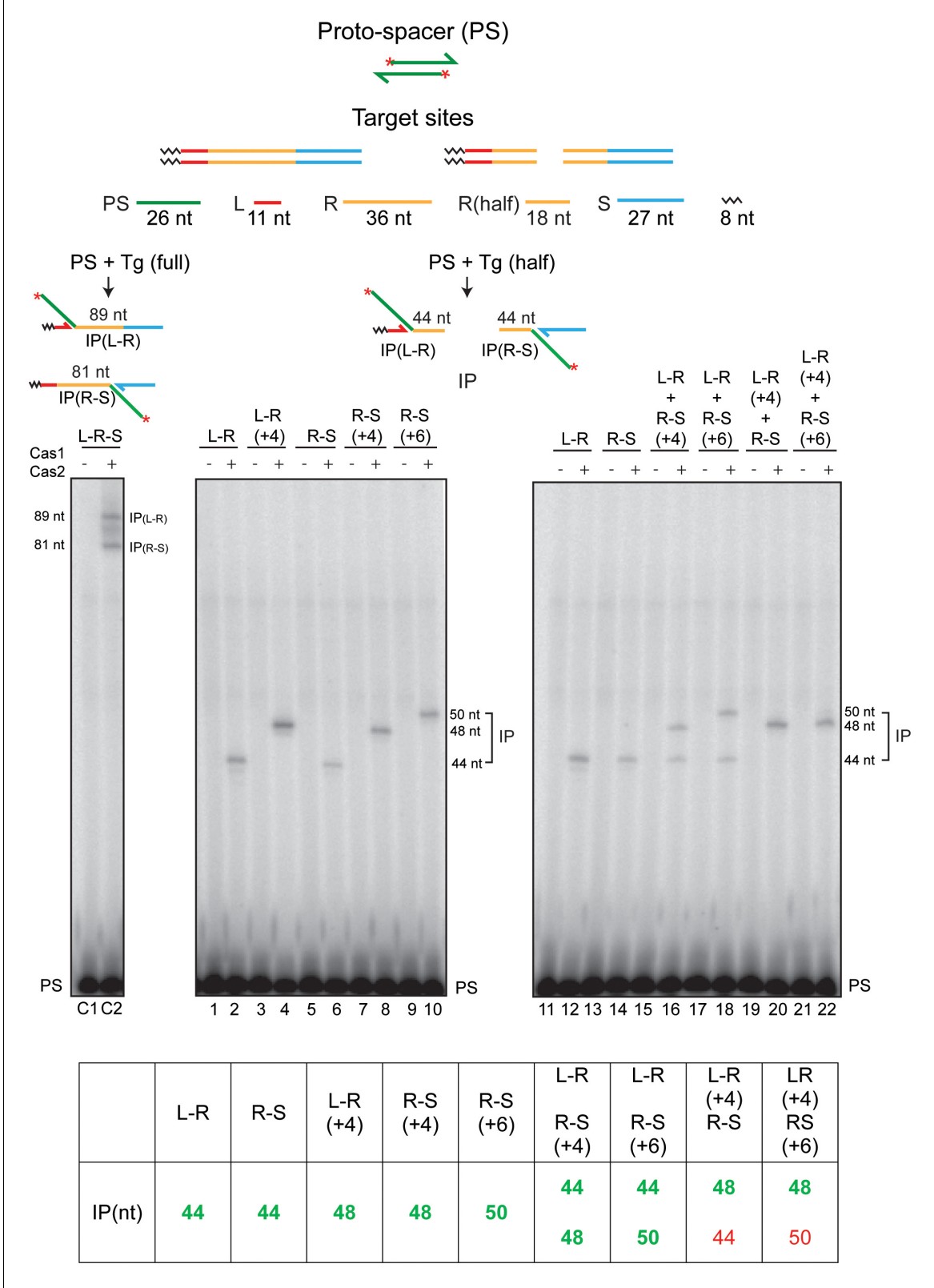

**Figure 4.** Cas1-Cas2 transfers the protospacer into linear half-target sites. The protospacer labeled with [32]P at its 5'-ends and the linear full- and half-target sites are diagrammed at the top. The minimal L-R or R-S half-site contained 18 bp of the L-proximal or R-proximal repeat sequence, respectively (*Figure 4—figure supplement 1*). Each half-site is the product of splitting a full-site into two exactly at the center of the 36 bp repeat. Reactions contained the labeled protospacer together with the indicated minimal L-R or R-S half-sites, or half-sites containing longer repeat sequences (+4 or +6

*Figure 4 continued*

bp) (*Figure 4—figure supplement 1*), as well as combinations thereof. Reactions were analyzed as described in *Figures 2* and *3*. The lanes C1 and C2 denote the control reaction with a normal target site containing the full-length repeat. The predicted integration products from the individual and mixed half-target site reactions are listed at the bottom. The experimentally observed products are distinguished from the undetected ones by the bold green and normal red fonts, respectively. IP = integration product; PS = protospacer.

The online version of this article includes the following figure supplement(s) for figure 4:

**Figure supplement 1.** Half-target sites used for protospacer integration assays.

the absence of linkage. The extent of repeat sequence present adjacent to the L-R junction and/or specific repeat sequences distal to this junction may contribute to the priority in target site recognition and order of strand transfer during spacer acquisition. It is possible that the integration reaction occurs within a Cas1-Cas2 complex harboring a non-covalently dimerized pair of half-sites, $(L-R)_2$, $(R-S)_2$, or (L-R)-(R-S), and the L-R(+4) dominance is manifested in this context.

## Apparent integration-disintegration antagonism during spacer acquisition

'Cut-and-paste' transposition during integration involves an unusually short transposon (the protospacer) and a particularly long target site (the CRISPR repeat). Their nearly matched sizes may partly relieve the steric impediments to insertion posed by DNA persistence length. Nevertheless, the strain within the strand transfer intermediate could potentially trigger disintegration mediated by the 3'-hydroxyl exposed at the leader or spacer end. Disintegration may offer a protective mechanism to prevent off-target integration events that could imperil the integrity of the bacterial genome (*Wright and Doudna, 2016*). The conformational dynamics of authentic integration at the CRISPR locus may displace the 3'-hydroxyl from the protospacer-repeat junction or misorient it with respect to the scissile phosphate. We investigated how disintegration might influence the second step of protospacer integration by taking advantage of substrates that mimic a semi-integrated protospacer.

The reactions were carried out in substrates assembled by hybridizing four oligonucleotides to represent semi-integration at the L-R junction (*Figure 5A*). The 3'-hydroxyl of the unintegrated protospacer strand would serve as the nucleophile for the second integration step. Similarly, the 3'-hydroxyl of the 'leader strand', and potentially water, could provide the nucleophiles for the disintegration reaction. When the 5'-end of the unintegrated protospacer strand was labeled, integration was observed at the R-S junction (81 nt band) (*Figure 5A*; lanes 2–6). The reaction was rapid and was nearly saturated within a minute under the conditions employed. Along with the 81 nt product, a smaller amount of the companion 89 nt product was also formed, whose yield showed a linear increase with time (*Figure 5A*; lanes 2–6). Formation of this minor product suggests that the phosphodiester bond at the L-R junction was restored by 3'-hydroxyl-mediated disintegration in at least a subpopulation of the starting substrate molecules. This junction could then serve as the target site for strand insertion from a disintegrated protospacer. Consistent with this interpretation, labeling the 5'-end of the leader strand revealed the 92 nt product expected for disintegration carried out by the leader 3'-hydroxyl to re-form the L-R junction (*Figure 5A*; lanes 2'−6'). Strikingly, disintegration was more prominent than integration.

An unexpected product that migrates slightly slower than the 81 nt band was observed in reactions containing the 5'-end label on the leader strand (*Figure 5A*; lanes 2'−6'). This product, which showed a steady increase with time, suggests the potential utilization of the leader 3'-hydroxyl by Cas1-Cas2 in an integration-like reaction (*Figure 5—figure supplement 1* and associated Supplementary Text S5). This uncharacterized product has no direct bearing on the possible functional link between integration and disintegration (see below and under 'Discussion').

In a substrate mimicking semi-integration at the R-S junction, Cas1-Cas2-promoted the second integration step at the L-R junction, yielding the expected 89 nt product (*Figure 5—figure supplement 1*). Resealing of the R-S junction via disintegration mediated by the 3'-spacer hydroxyl was also evident from the smaller amounts of the 81 nt product formed by the semi-integration of a released protospacer at R-S. Unlike the first strand transfer step, which obeys a strict order during protospacer insertion, the second strand transfer (and subsequent strand transfers) may proceed

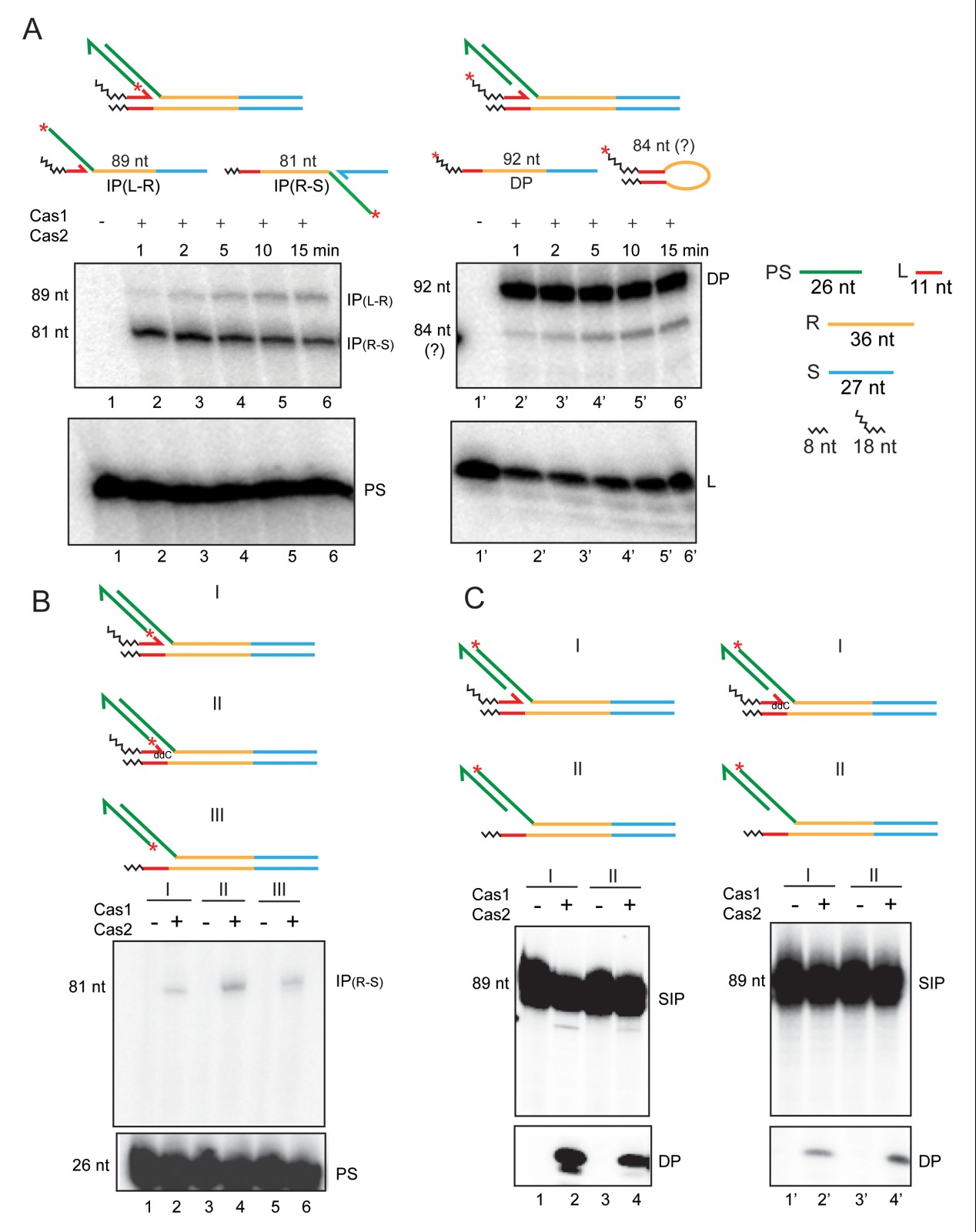

**Figure 5.** Second integration and disintegration occur in substrates mimicking a semi-integrated protospacer. (A) Schematics of semi-integrated protospacers at the L-R junction (top row) and the possible products formed from them by Cas1-Cas2 action (bottom row) are shown. The [32]P-label was placed at the 5′-end of the unintegrated protospacer strand or the leader strand to follow the second integration and disintegration events, respectively. (B) Integration of the [32]P-labeled protospacer strand at the R-S junction was monitored in the indicated substrates. (C) Disintegration was

*Figure 5 continued on next page*

*Figure 5 continued*

assayed in the same substrates as in (B), except that the [32]P-label was placed at the 5′-end of the integrated protospacer strand. ddC = dideoxy C; DP = disintegration product; IP = integration product; PS = protospacer; SIP = semi-integrated protospacer.

The online version of this article includes the following figure supplement(s) for figure 5:

**Figure supplement 1.** Cas1-Cas2 mediates integration and disintegration of a semi-integrated protospacer.

from L-R to R-S or R-S to L-R with roughly equal efficiency. The prominence of disintegration (DP; 92 nt) and the rapid kinetics of disintegration and the second integration at the R-S junction (IP$_{(R-S)}$; 81 nt) (*Figure 5A*) suggest that the two events are potentially coupled.

## Water-mediated disintegration of a semi-integrated protospacer

Disintegration from one of the semi-insertion sites (the L-R or the R-S junction) could produce a protospacer that is either free in solution or is concomitantly inserted at the other site. The coupling of integration to disintegration, consistent with their rapid saturation kinetics (*Figure 5A*), could be promoted by the conformational strain associated with orienting the reactive 3′-hydroxyl for the second transfer step. We wished to test whether preventing the disintegration reaction at the L-R junction in the semi-integrated state would reduce integration at the R-S junction.

We attempted to block the disintegration competence of the leader strand in the substrate with the protospacer semi-integrated at the L-R junction by placing a terminal 3′-dideoxy nucleotide, or by removing this strand altogether. Neither of these substrate designs eliminated or reduced the extent of integration at the R-S junction (81 nt band in *Figure 5B*). However, labeling the 5′-end of the protospacer strand integrated at the L-R junction revealed that disintegration occurred even in the absence of the leader strand (*Figure 5C*).

Water-mediated disintegration in the semi-integrated substrate could potentially assist the integration of the second protospacer strand in a coupled fashion. Disintegration efficiency of water alone (in the absence of the leader strand) was ~40% of that of the 3′-hydroxyl and water combined (in the presence of the leader strand) (*Figure 5C*, compare lanes 2 and 4). When the leader strand carried a di-deoxy 3′-end, there was some reduction (to ~70%) in water-mediated disintegration (*Figure 5C*; compare lanes 2′ and 4′), presumably due to steric interference. Such interference could be mitigated if, during a standard reaction, protospacer integration at the L-R junction is accompanied by the displacement or misalignment of the leader 3′-hydroxyl. We note that water is ~50% as competent a nucleophile as the 3′-hydroxyl for disintegration.

The 3′-hydroxyl-mediated and water-mediated disintegration reactions differ in that the former rejoins L to R (or R to S) whereas the latter leaves open the strand nick between L and R (or R and S). Processing of such a nick by DNA repair enzymes has functional implications in protospacer acquisition (see Discussion).

## Coupling between integration of the second strand and disintegration of the first

To better appreciate the extent of coupling between integration and disintegration, we modified the design of the semi-integrated state. The rationale was to join the integrated and unintegrated protospacer strands, so that the fate of both could be followed simultaneously.

In the 'trombone' substrates for testing coupled integration-disintegration (*Figure 6*; see also *Figure 6—figure supplement 1*), one protospacer strand was integrated at the L-R junction. Furthermore, the 3′-end of the spacer top strand was tethered to the 5′-end of the unintegrated protospacer strand. Our reasoning that the 30 nt long flexible single-stranded tether (5′T$_4$(CT)$_{10}$T$_6$3′; ~240 Å) should have no (or only minimal) effect on Cas1-Cas2 activity was upheld by experiments. The spacer in these substrates was truncated to contain only 10 bp abutting the repeat. The shortened spacer had no adverse effect on strand transfer of an untethered protospacer from the L-R to the R-S junction (*Figure 6A*; lanes 2 and 4).

The predicted products of the second integration event in a trombone substrate would differ by 26 nt (the size of the integrated protospacer strand), depending on whether it is associated with disintegration or not. Labeling the 5′-end of the protospacer strand integrated at L-R can only reveal integration at R-S without disintegration at L-R. The size of this product analyzed on a denaturing

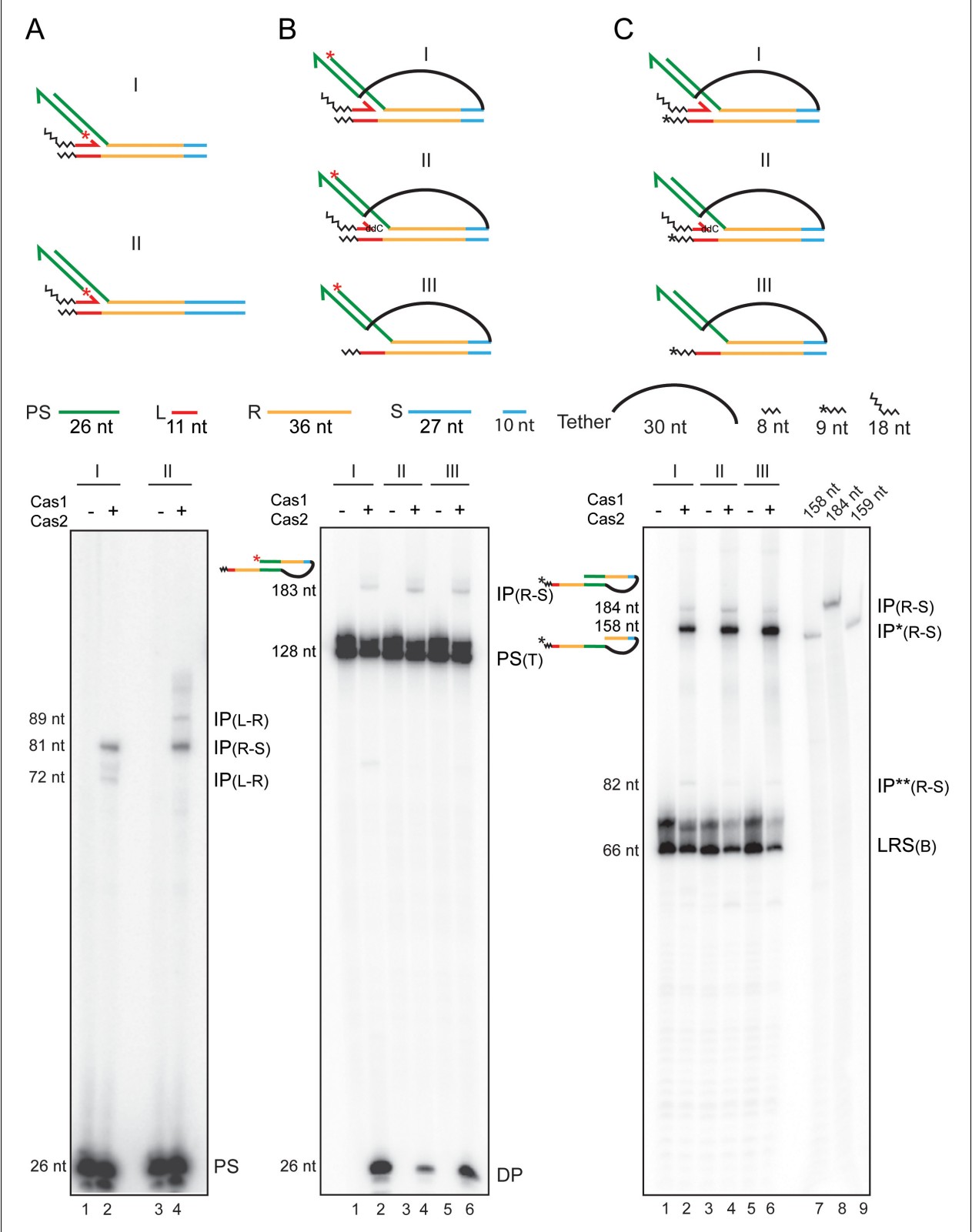

**Figure 6.** Second strand integration by Cas1-Cas2 occurs in concert with first-strand disintegration. Schematics of the substrates used for the individual sets of assays (**A–C**) are placed above the respective gel panels. The layout of the trombone substrates (**B, C**) from a perspective of the Cas1-Cas2-protospacer structure is shown in *Figure 6—figure supplement 1*. (**A**) Integration reactions were performed in the two substrates that differed only in their spacer lengths. The truncated spacer (10 bp long) lacked the 17 bp distal to the R-S junction. (**B, C**) The substrate configurations were similar for

*Figure 6 continued on next page*

*Figure 6 continued*

the two reaction sets, except for the position of the $^{32}$P label. In (**B**), the integrated protospacer strand carried the label at the 5′-end (red asterisk). In (**C**), one labeled nucleotide (shown by the asterisk) was added to the 3′-end of the bottom strand. DP = disintegration product; IP = integration product; IP* = integration product coupled to disintegration; IP** = integration product uncoupled from (but preceded by) disintegration; PS = protospacer; PS (T) = protospacer (tethered); LRS (B) = Substrate bottom strand. Lanes 7–9 in (**C**), oligonucleotide length markers.

The online version of this article includes the following figure supplement(s) for figure 6:

**Figure supplement 1.** Structural perspective of a trombone substrate in which an integrated protospacer strand is linked to the unintegrated strand by a single-stranded tether.

**Figure supplement 2.** Distinct products formed by Cas1-Cas2 action on trombone substrates.

gel will be 183 nt (26 nt + 26 nt protospacer + 36 nt + 36 nt repeat + 10 nt spacer + 30 nt tether + 11 nt leader + 8 nt extension to leader). If integration at R-S were to be accompanied by disintegration at L-R, the predicted 157 nt product would be invisible as the release of the spacer strand (26 nt) results in the loss of the radioactive label as well. Labeling the 3′-end of the bottom strand containing the R-S junction target site (by adding one α-$^{32}$P-labeled nucleotide in a Klenow polymerase reaction) would reveal integration events coupled to disintegration (158 nt) or uncoupled from it (184 nt). Detailed schematic illustrations of the possible products from the 5′- and 3′-end-labeled substrates and their predicted sizes are given in *Figure 6—figure supplement 2*.

Formation of the 5′-labeled 183 nt product (signifying no disintegration) was weak, though detectable, for the three substrates tested (*Figure 6B*; lanes 2, 4, and 6; *Figure 6—figure supplement 2A*). Two of these contained a leader strand (one with a hydroxyl and the other with a deoxy 3′-end), and the third lacked this strand. At the same time, disintegration mediated by the leader 3′-hydroxyl or by water was readily observed (26 nt; *Figure 6B*; lanes 2, 4, and 6; *Figure 6—figure supplement 2A*). For all three substrates labeled at the 3′-end, the shorter product (158 nt) exceeded the longer one (184 nt) (*Figure 6C*; lanes 2, 4, and 6; *Figure 6—figure supplement 2B*). The difference was eight- to tenfold when the leader strand carried a 3′-hydroxyl (*Figure 6C*; lane 2) or had a deoxy 3′-end (*Figure 6C*; lane 4), and >15-fold when the leader strand was absent (*Figure 6C*; lane 6). The maximum yield of the 158 nt product in the absence of a leader strand suggests that the second integration is more efficient when disintegration is performed by water rather than by the 3′-hydroxyl.

Several features of the protospacer strand transfer steps are revealed by the trombone substrates (*Figure 6B,C*; see also *Figure 6—figure supplement 2*). First, integration of the second strand without disintegration of the first is quite infrequent. Second, disintegration of a protospacer from the L-R junction followed by its integration at the R-S junction in a two-step process is also rare. The IP (R-S)** band (82 nt; *Figure 6C*) is the result of such a reaction, namely, insertion of the 26 nt protospacer strand released from the L-R junction into the R-S junction. Thus, integration of the second strand and disintegration of the first are most often concerted events, and two-ended insertion events by the Cas1-Cas2 complex seldom occur in vitro. Finally, water being better at promoting disintegration coupled integration than the 3′-hydroxyl is significant. The strand nick left unsealed during the water-mediated reaction would permit seemingly abortive intermediates to be processed by bacterial DNA repair systems into successful spacer insertion (see Discussion). Thus, paradoxically, disintegration may be an important (but unappreciated) contributing factor in the adaption stage of CRISPR immunity.

## Discussion

Cas1-Cas2-mediated adaptation is strongly conserved across CRISPR systems, but is less understood than the much more diverse interference mechanisms (*Wiedenheft et al., 2012*; *Krupovic et al., 2017*; *Yosef et al., 2012*; *Jiang and Doudna, 2015*; *Mohanraju et al., 2016*). Our analysis indicates that the integration by Cas1-Cas2 in vitro starts with the insertion of one protospacer strand at the L-R junction, and utilizes a ruler-like mechanism to attempt integration of the second strand at the R-S junction. However, a successful second integration event is most often associated with the disintegration of the first one. We propose below plausible models for disintegration-promoted spacer acquisition that are accommodated by well-established DNA transposition mechanisms.

Two-ended protospacer integration, without intervening disintegration, is conceivable in vivo. A high-order adaptation complex in which Cas1-Cas2 is assisted by additional Cas proteins (e.g., Cas4, Csn2, and Cascade/Cas9) and by bacterial factors (e.g., RecBCD and AddAb) (*Amitai and Sorek, 2016*; *Nuñez et al., 2015a*; *Yosef et al., 2012*; *Heler et al., 2015*; *Wei et al., 2015a*; *Levy et al., 2015*) may deter disintegration by coupling DNA processing to strand transfer without protospacer dissociation. Dynamic remodeling of the complex in conformation and/or subunit stoichiometry, as suggested by Cas1-Cas2-Csn2 cryo-EM structures (*Wilkinson et al., 2019*), could facilitate the individual reaction steps. However, if a pre-processed protospacer were to escape from the complex, it is still able to complete Cas1-Cas2-catalyzed, and disintegration-promoted, integration.

## From invading DNA to spacer deposition at the CRISPR locus: Variations of a shared DNA transposition theme

An invading virus/plasmid DNA is processed by bacterial nuclease/helicase complexes into smaller sized fragments (pre-spacers), to be further processed by Cas proteins into protospacers suitable for integration at the CRISPR locus (*Amitai and Sorek, 2016*; *Levy et al., 2015*; *Kim et al., 2020*; *Drabavicius et al., 2018*; *Kieper et al., 2018*; *Lee et al., 2018*; *Rollie et al., 2018*; *Shiimori et al., 2018*). The early and subsequent processing events may be coordinated by communication between the Cas proteins and nuclease/helicase proteins. In principle, the action of Cas1-Cas2 (perhaps with the assistance of partner Cas proteins) directly on the invading foreign DNA, or on shorter processed fragments generated from it, may give rise to unexcised protospacers with cleaved single-stranded ends or fully excised free-standing protospacers, both of which would be competent to initiate the integration reaction (*Figure 7*). These situations would be analogous to replicative and cut-and-paste DNA transposition, respectively (*Haniford and Chaconas, 1992*; *Haniford and Ellis, 2015*; *Harshey, 2014*). In the cut-and-paste mechanism, only limited gap repair DNA synthesis is required to re-establish the native organization of the CRISPR locus, but with the newly acquired spacer (*Figure 7*). In the replicative mechanism, DNA synthesis is more extensive, and the intermediate is resolved via recombination to yield the same end product as in cut-and-paste transposition (*Figure 7*). Variations of these themes, combining aspects of replicative and cut-and-paste transpositions may also be envisaged (*Figure 7—figure supplement 1*).

## Orientation specificity in protospacer integration

Spacer acquisition exclusively in its functional orientation, that is, directed transfer of the DNA end resulting from strand cleavage at PAM to the L-R junction, is easier to imagine in a high-order Cas protein complex capable of both pre-spacer processing and protospacer insertion than in a Cas1-Cas2 complex associated with an already processed protospacer. In vitro, either of the two protospacer strands is equally capable of integration, and coupled cleavage transfer is not detectable in DNA substrates that mimic PAM-containing pre-spacers (*Figure 2*; *Figure 2—figure supplement 3*). Retention of spacer function even in the absence of orientation specificity in strand transfer is theoretically possible by dual PAM recognition (Supplementary Text S3 in *Figure 2—figure supplement 3*).

## Cas1-Cas2-mediated integration of a pre-processed protospacer: Assistance from disintegration

If 'normal' (two-ended) integration by the adaptation complex (*Figure 1A*; *Figure 7*) is disrupted after only one protospacer strand has been transferred to the target, salvage pathways may promote the maturation of such intermediates into complete spacer insertions. These pathways may also come into play during the integration of a protospacer that is dissociated from the adaptation complex and is captured by the Cas1-Cas2 complex. Integration under this scenario, which mimics the in vitro system, is expected to be predominantly one-ended. We suggest that disintegration events, which become prominent during the transfer of the second strand, play a central role in the salvage mechanisms.

Water-mediated disintegration of the protospacer at the L-R junction during strand transfer to the R-S junction will produce an intermediate with a gap in the bottom strand and a gap plus nick in the top strand (*Figure 8*; left). An analogous intermediate with the gap plus nick in the bottom strand may also be formed (*Figure 8*; right). Here, a protospacer is first transferred to the R-S

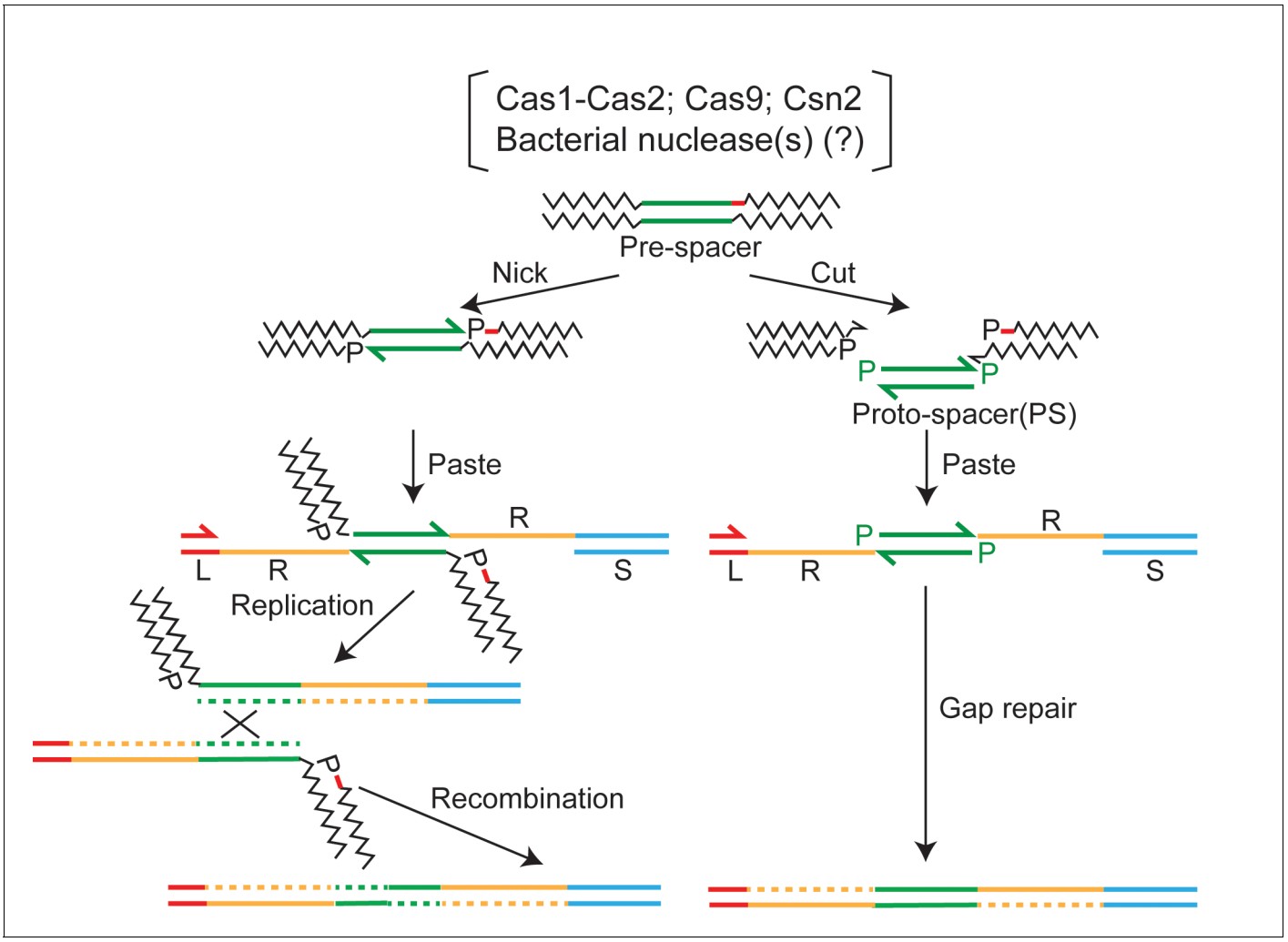

**Figure 7.** Two-ended integration of the protospacer by the spacer acquisition complex. Two possible schemes for protospacer integration are outlined. The pathway shown at the left is equivalent to replicative DNA transposition. The pre-spacer is nicked at the protospacer 3′-ends, at least one cleavage being adjacent to a PAM sequence (shown by the red dash). The 3′-hydroxyls are then transferred to the leader-repeat (L–R) and repeat-spacer (R–S) junctions of the target site. Replication across L and PS (protospacer) followed by recombination within the duplicated PS completes integration. An alternative mode of processing the strand transfer intermediate involves removal of the pre-spacer DNA flanking PS, and gap repair by limited DNA synthesis and ligation (*Figure 7—figure supplement 1*). The integration pathway shown at the right follows cut-and-paste DNA transposition. The steps involve excision of PS from the pre-spacer, strand transfer to L-R and R-S junctions, gap repair, and ligation. In either pathway (replicative or cut-and-paste), directing the PAM-adjacent 3′-hydroxyl to the L-R junction will ensure the functional orientation of the inserted spacer. PAM, protospacer adjacent motif.

The online version of this article includes the following figure supplement(s) for figure 7:

**Figure supplement 1.** Spacer acquisition by a mechanism that combines aspects of replicative and cut-and-paste transposition mechanisms.

junction with concomitant 3′-hydroxyl-mediated disintegration from the L-R junction (and restoration of this junction). This is followed by a subsequent integration in the reverse direction, with disintegration at the R-S junction promoted by water. The spacer insertion event can be completed by gap-filling/displacement DNA synthesis followed by ligation (*Figure 8*). Thus, disintegration reactions mediated by the 3′-hydroxyl and by water are both relevant, though in mechanistically distinct ways, to the rescue of semi-integrated protospacers into fully integrated spacers.

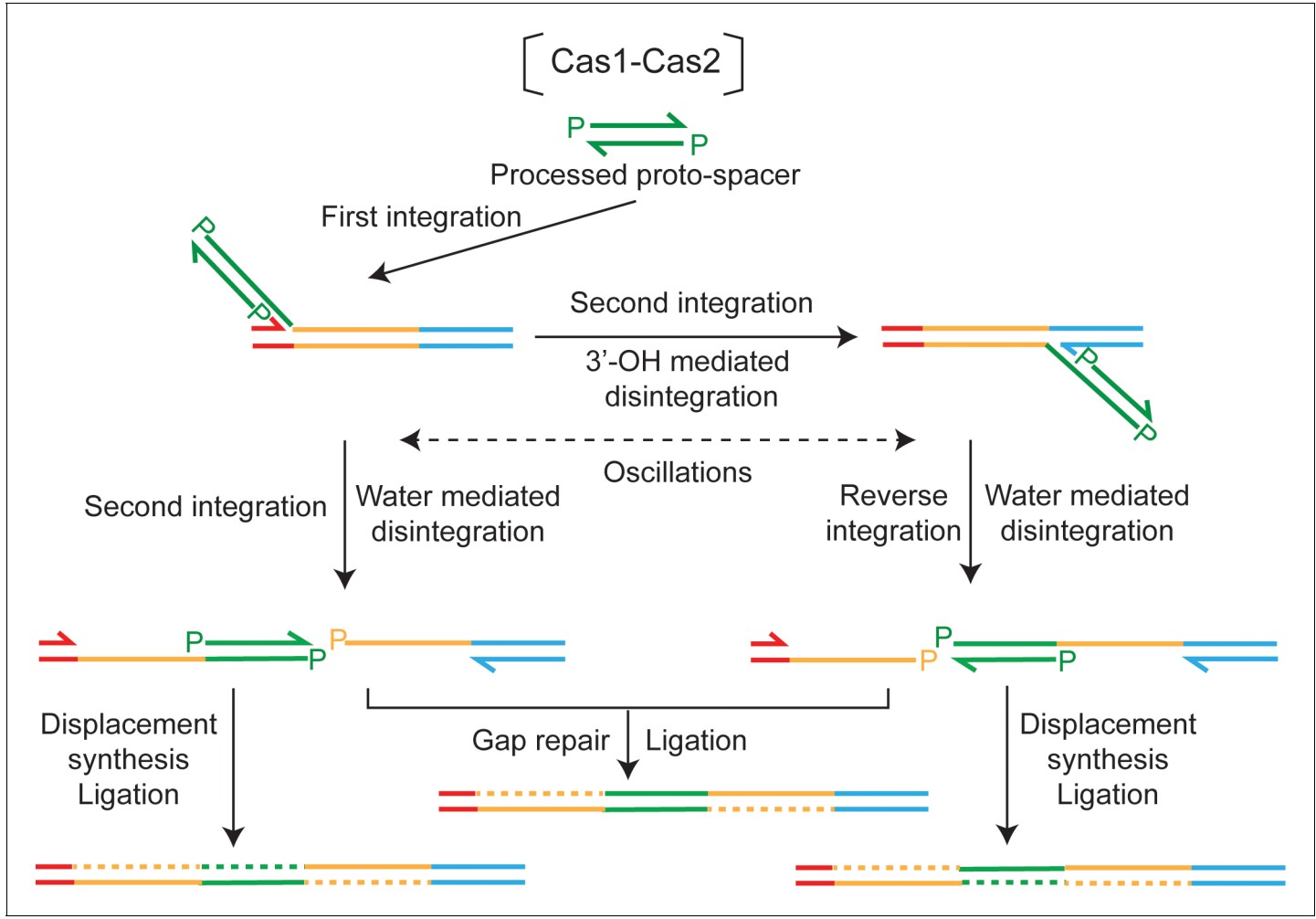

**Figure 8.** A model for disintegration-assisted integration. Integration of the protospacer by Cas1-Cas2 is predominantly one-ended. Following the initial strand transfer to the L-R junction, transfer of the second strand to the R-S junction is associated with disintegration at L-R. Oscillations of the semi-integrated protospacer become possible when disintegration mediated by the 3'-hydroxyl re-forms the L-R or R-S junction at the disintegration site. Disintegration mediated by water during a second strand transfer step will block oscillations. However, the nick between L and R or R and S will be preserved, and can be utilized by DNA repair machineries to complete protospacer insertion. Formation of the repair-suitable intermediate shown at the left is dependent only on water-mediated disintegration, the 3'-hydroxyl-mediated disintegration being dispensable. However, both types of disintegration are required to generate the intermediate shown at the right.

## Disintegration relieves topological stress during protospacer integration

Disintegration is generally viewed as antithetical to the completion of the two-strand transfer events during transposition reactions, which are analogous to protospacer insertion at the CRISPR locus. However, the in vitro Cas1-Cas2 reactions suggest that disintegration can generate DNA ends capable of triggering repair events that convert one-ended strand transfer intermediates into normal insertion products. This mechanism may be especially important when two-ended insertion is impeded by the transposon being considerably shorter than the persistence length of DNA. The strand transfer complex (or post-synaptic complex; PSC) in this case would be under considerable torsional stress. A recent single-molecule FRET analysis suggests that the PSC formed during protospacer integration may be resolved by transcription from the CRISPR leader accompanied by limited DNA replication (*Budhathoki et al., 2020*). Resolution may also be achieved by replication, although it is argued to be less likely. The lag time in the arrival of a replication fork at the PSC in a slow-growing bacterium may cause disintegration of the protospacer and its potential reintegration in the non-functional orientation. Replication through the unwound repeat and protospacer would produce

undesirable double-strand DNA breaks. However, if the integration complex functions as a strand transfer-DNA repair machinery, the DNA ends may remain protected within it to be resolved rapidly (*Figure 7*; pathway at the left). Precedent exists for the involvement of the replisome in the repair of the strand transfer intermediate formed during the insertion of infecting phage Mu DNA into the *E. coli* chromosome (*Jang and Harshey, 2015*).

The present study demonstrates that an alternative solution to transcription/replication in simplifying PSC topology is the hydrolytic nicking of DNA (disintegration), allowing transposition (integration) to be completed by ensuing DNA repair. An analogy can be drawn to the topological problem of DNA replication. Here, the torsional stress built up ahead of the fork is removed by topoisomerases cutting DNA strands and then resealing them to permit continued fork progression. The transesterification chemistry utilized by these enzymes (*Reece and Maxwell, 1991*; *Vos et al., 2011*) ensures strand cutting and repair in rapid succession without net DNA loss or gain. The distinct mechanisms for topology simplification utilized by DNA replication and CRISPR adaptation machineries are consistent with their contrasting biological purposes—maintaining genome integrity versus promoting an orderly genome rearrangement.

### A potential broader role for disintegration in DNA transposition

In a general sense, disintegration may not be limited to spacer acquisition by CRISPR loci. It may assist the movement of mini-transposons or short orphan transposons that are mobilized by ectopically supplied transposases. Disintegration need not always be counterproductive even during normal transposition, provided the DNA repair machinery is able to recommence the interrupted reaction. The Cas1-Cas2 system highlights the resourcefulness of biological catalysis in repurposing an apparently abortive reaction toward a physiologically meaningful outcome.

## Materials and methods

### Plasmids

Protein expression plasmids and substrate plasmids used for in vitro assays were kindly provided by the Doudna laboratory (*Wright and Doudna, 2016*).

### Oligonucleotides

Oligonucleotides used for the assembly of protospacers and target sites were purchased from IDT (listed in *Supplementary file 1*). Oligonucleotides >100 bp in length were synthesized as two separate chains and were ligated with the assistance of a short bridging splinter oligonucleotide. The 5′-end was made competent for ligation by phosphorylating it with ATP using T4 polynucleotide kinase.

Hybridizations were performed by mixing the requisite oligonucleotides in 20 mM HEPES-NaOH (pH 7.5), 25 mM KCl, and 10 mM MgCl$_2$, immersing the tube containing the mixture in a water bath at 90℃ for 5 min, and letting the bath cool slowly to room temperature. After adding DTT (dithiothreitol) and DMSO (dimethyl sulfoxide) to final concentrations of 1 mM and 10%, respectively, they were immediately used for reactions or stored at −20℃ until use (not more than a week).

### Proteins

His6-tagged proteins, expressed individually as MBP (maltose-binding protein) fusions in *E. coli*, were purified by affinity chromatography over Ni-sepharose using standard protocols. When desired, MBP was removed by cleavage at a TEV protease target site placed at the fusion junction. MBP-free proteins were purified by Ni-affinity chromatography. Final protein preparations were dialyzed against 20 mM Tris-HCl (pH 7.5), 150 mM KCl (Cas1, Csn2, and Cas9) or 500 mM KCl (Cas2), 1 mM TCEP (Tris(2-carboxyethy)phosphine hydrochloride), and stored at −70℃ after addition of 10% glycerol (v/v). There was no difference in the activities between the MBP-fusion and MBP-cleaved forms of Cas1 and Cas2. All data shown here were obtained using the Cas1 and Cas2 fusion proteins. Cas9 and Csn2 were used after the removal of MBP.

### Radioactive labeling

5′-end labeling of oligonucleotides was carried out using γ-$^{32}$P-labeled ATP and T4 DNA ligase. For labeling the 3′-end, the Klenow fill-in reaction was performed in the presence of α-$^{32}$P-labeled dATP. A short-oligonucleotide with a 5′ T overhang hybridized to the oligonucleotide being labeled ensured the incorporation of one-labeled A.

### Cas1-Cas2 activities

Reactions were performed in 20 mM HEPES-NaOH (pH 7.5), 25 mM KCl, 10 mM MgCl$_2$, I mM DTT, and 10% DMSO (*Wright and Doudna, 2016*) containing 5% PEG (polyethylene glycol 1500; MilliporeSigma). Incubations were carried out in 20 μl reaction mixtures at 37°C for 1 hr in most cases. In time course analyses, incubation periods ranged from 1 min to 2 hr, reactions were stopped by addition of SDS (0.2% final; w/v) and heating to 95°C for 5 min. Samples were analyzed by electrophoresis in 1.5% agarose gels or 12% denaturing polyacrylamide-urea gels (19:1 crosslinking). In one set of assays probing plasmid topoisomer distributions, 0.4 μg chloroquine/ml was present in the agarose gel and the running buffer. DNA bands were visualized by ethidium bromide staining, fluorography, or by phosphorimaging depending on the particular assays.

Standard reactions utilizing supercoiled plasmids contained ~10 nM plasmid DNA, ~34 nM Cas1, and ~17 nM Cas2 (2:1 molar ratio of Cas1:Cas2). In reactions containing a mixture of Cas1 and Cas1 (H205A) (dCas1), the combined concentration of the two per reaction was kept as ~34 nM. For reactions utilizing $^{32}$P-labeled oligonucleotides, the DNA substrate was held at 25 nM with Cas1 and Cas2 at ~68 nM and 34 nM, respectively. Proteins were diluted from stock aliquots to a concentration of 1 μM in the reaction buffer. Cas1-Cas2 were mixed in the requisite molar ratios and kept on ice for 10 min before addition to reaction mixtures. For some reactions, Cas1-Cas2 pre-incubation on ice included Csn2 or Cas9 or both. The stoichiometry of the additional proteins or deviation from the standard 2:1 molar ratio of Cas1:Cas2 is indicated in the pertinent figures.

### Topoisomerase reactions

A titrated amount of *E. coli* topoisomerase I was used to obtain partial relaxation of plasmid DNA. Reactions were carried out in the supplier (New England Biolabs) recommended buffer in 30 μl volumes containing 300 ng supercoiled plasmid DNA per tube. After incubation at 37°C for varying times, reactions were quenched with SDS (0.2% final; w/v) and heated to 95°C for 3 min before electrophoresis in 1.5% agarose.

### Ligation of nicked plasmid DNA

Plasmid DNA, nicked at a single site using Nb.BtsI (New England Biolabs), was ligated using T4 DNA ligase and the ligation buffer (obtained from the supplier). Each reaction mixture containing 300 ng nicked plasmid and 200U ligase was incubated for 1 hr at 37°C before inactivating the enzyme by SDS addition (0.2% final; w/v) and heating to 95°C for 3 min. Samples were analyzed by electrophoresis in 1.5% agarose.

### Quantification of reaction efficiencies

$^{32}$P-labeled bands on polyacrylamide gels were detected using a storage phosphor screen (Bio-Rad) at different exposure times to optimize signal detection without saturation of the screen. Scanning was performed using a Typhoon Trio phosphorimager (GE Healthcare), and image analysis was performed using the software Quantity One (Life Sciences Research) and ImageJ (NIH).

## Acknowledgements

The authors thank Drs. Addison Wright and Jennifer Doudna for Cas1-Cas2 plasmids and members of the Finkelstein and Jayaram labs for helpful discussions. The work was supported by NIGMS R01GM124141 (to IJF), NSF MCB-1049925 and MCB-1949821 (to MJ), and the Welch Foundation grants F-1274 (to MJ) and F-1808 (to IJF).

## Additional information

### Funding

| Funder | Grant reference number | Author |
|---|---|---|
| National Institute of General Medical Sciences | R01GM124141 | Ilya J Finkelstein |
| National Science Foundation | MCB-1049925 | Makkuni Jayaram |
| National Science Foundation | MCB-1949821 | Makkuni Jayaram |
| Welch Foundation | F-1274 | Makkuni Jayaram |
| Welch Foundation | F-1808 | Ilya J Finkelstein |

The funders had no role in study design, data collection and interpretation, or the decision to submit the work for publication.

### Author contributions

Chien-Hui Ma, Conceptualization, Data curation, Formal analysis, Validation, Investigation, Visualization, Methodology, Writing - review and editing; Kamyab Javanmardi, Conceptualization, Investigation, Visualization, Writing - review and editing; Ilya J Finkelstein, Conceptualization, Resources, Formal analysis, Supervision, Funding acquisition, Validation, Visualization, Methodology, Writing - original draft, Project administration, Writing - review and editing; Makkuni Jayaram, Conceptualization, Resources, Formal analysis, Supervision, Funding acquisition, Validation, Methodology, Writing - original draft, Project administration, Writing - review and editing

### Author ORCIDs

Kamyab Javanmardi (iD) http://orcid.org/0000-0002-6449-6709
Ilya J Finkelstein (iD) https://orcid.org/0000-0002-9371-2431
Makkuni Jayaram (iD) https://orcid.org/0000-0001-9640-6264

### Decision letter and Author response

Decision letter https://doi.org/10.7554/eLife.65763.sa1
Author response https://doi.org/10.7554/eLife.65763.sa2

## Additional files

### Supplementary files

• Source data 1. Set 1 of source data files.

• Source data 2. Set 2 of source data files.

• Supplementary file 1. List of oligonucleotides.

• Transparent reporting form

### Data availability

The data generated and analyzed in this study are provided in the manuscript and supplementary data.

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
