## [Decision Letter]

**Acceptance summary:**

A key step in the acquisition of viral immunity in bacteria by CRISPR systems is the integration of foreign DNA sequences into the host genome. Integration is carried out by a dedicated transposase encoded in the CRISPR array. The current work is of note for highlighting the potential for water-mediated disintegration reactions – events where the integration of phage DNA into a target site is reversal – in controlling how DNA is inserted into the chromosome, an event that may be functionally important in the adaptive phase of CRISPR-mediated bacterial immunity.

**Decision letter after peer review:**

[Editors’ note: the authors submitted for reconsideration following the decision after peer review. What follows is the decision letter after the first round of review.]

Thank you for submitting your work entitled "Disintegration promotes proto-spacer integration by the Cas1-Cas2 complex" for consideration by *eLife*. Your article has been reviewed by 2 peer reviewers, and the evaluation has been overseen by a Reviewing Editor and a Senior Editor. The reviewers have opted to remain anonymous.

We are sorry to say that, after consultation with the reviewers, whose evaluation is detailed below, we have decided that your work will not be considered further for publication by *eLife*.

Summary:

In this manuscript, an in vitro Cas1-Cas2 model system is used to study the reaction used to insert foreign DNA elements into a CRISPR array during the adaptive phase of immunity. The authors propose that hydrolysis of one end of the transposon DNA may be the primary mechanism for the insertion of very small DNA elements (which are difficult to bend tightly) that are found for the proto spacer sequences, and that cellular repair pathways are responsible for ligating the CRISPR array back together in vivo. The findings additionally suggest that water-mediated disintegration has an unappreciated role in the generation of CRISPR arrays as part of the bacterial immune response. These hypotheses are intriguing and of potential interest to those in the CRISPR field. However, it is unclear how this in vitro study, which does not monitor the full the reaction (directionality is lost due to the lack of a PAM sequence in the substrate and several required cellular factors are missing), relates to transposition as it occurs in vivo. Overall, this is an interesting study that challenges the current thinking in the field, but it does not present sufficient evidence to establish the physiological significance of the observed effects, thereby limiting its potential broader impact.*Reviewer #1:*

In this manuscript Ma and colleagues present a biochemical investigation into integration by the Spy Cas1-Cas2 integrase, a key component of the type II CRISPR immune response. From this work they challenge the view that disintegration (the reverse of integration) only functions as an off-pathway reaction (proof reading) and propose that disintegration-promoted integration is functionally important for integration into CRISPR arrays.

CRISPR integration proceeds through two steps. In the first, one strand of the prespacer (fragment of foreign DNA) is integrated at the leader-repeat (R-L) junction of the CRISPR array. In the second step, the other strand of the prespacer integrates at the repeat-spacer (R-S) junction resulting in a fully integrated prespacer. The authors propose that, in vitro, water-mediated disintegration at the first site (R-L) is concomitant with second site integration (R-S). The authors present good evidence that this is happening in vitro though a series of elegant biochemical experiments using plasmid and oligo substrates.

This is an intriguing hypothesis but its significance is unclear. As the authors themselves acknowledge it could very well be that disintegration is not required in vivo and that this is an in vitro feature of the reaction. Admittedly, demonstrating that disintegration is important in vivo would be a very difficult task. In addition, the in vitro system used here is only partially reconstituted. The substrates lack a PAM sequence, which is necessary for protospacers to be incorporated in the correct orientation and may help direct the first integration event to the L-R junction. Presumably because of this all the reactions presented do not analyze the orientation of the incorporated prespacer sequence. Cas9 and Csn2 are also absent (as are other potentially required host factors), which are necessary for correct integration in vivo. Any of these missing components could change the behavior of the system with respect to disintegration.

The authors present a very interesting hypothesis, disintegration is a key step in CRISPR adaptation, but do not test this hypothesis in a meaningful way. This limits the interest of this work for a broad audience.

In addition to the above I have the following comments/suggestions:

Figure 1: using a plasmid that lacks a CRISPR array is an important control. It is necessary to establish that the integration events being monitored are indeed at the CRISPR array because formation of half-sites at other locations is expected to result in disintegration. It could also be beneficial to include sequence data to map the site of integration in the pCRISPR targets. This has been done before in this system (PMID: 27595346) but would again bolster the conclusions presented here.

On page 19; the authors state that "…the kinetic correspondence between disintegration and the second integration reactions suggest that the two events are potentially coupled". However, the data presented in Figure 5A do not support this conclusion. The extent of second-site integration (the 81 nt band) and first site disintegration (the 92 nt band) do not change over the measured time frame. Both reactions appear to be over at 1 minute, which is also the first time point. Therefore it is impossible to know if both reactions occur at the same rate or if one is faster than the other.

With the exception of Figure 6, there are no markers, or sequence reaction products, on any of the acrylamide gels presented in the manuscript. It's therefore impossible for the reader to evaluate the size of any of the bands for themselves. This information should be included in any revised manuscript at least at the first instance of a particular sized product.

The experiments described in Figure S8 lack a Cas1 dead control. It's important to remove the possibility that the observed cleavage pattern results from a contaminating nuclease.

Some of the experiments are difficult to follow. For example it took me a long time to really understand the experiments in Figure 6, which are really elegant experiments but difficult to follow. I worry that this limits the accessibility of this work. I would encourage the authors to reevaluate how experiments are presented such that they are accessible to a broad audience. It would be helpful to not have to refer to a supplemental figure to understand the main figure i.e. the main figures should be self-contained.

*Reviewer #2:*

Ma and colleagues have show that when DNA fragments are inserted into a CRISPR array in vitro, single end integrations are the primary products of proto-spacer insertion and double-end insertions are rare. The reason for this outcome is that the second insertion event is accompanied by 3'-OH mediated disintegration or hydrolysis at the initial insertion site. A double-strand break would remain following filling of the repeat sequence gaps in vivo in the case of the hydrolysis products. The authors propose that hydrolysis of one end of the transposon DNA may be the primary mechanism for insertion of very small DNA elements (which will be difficult to bend tightly) as found for the proto spacer sequences and that cellular repair pathways would be responsible for ligating the CRISPR array back together.

The manuscript is well written and the experiments carefully planned. The findings will be of interest to researchers studying transposition-like mechanisms in a broad range of systems. The experiments focus on analyses of Cas1-Cas2 catalyzed in vitro reactions using cleverly designed DNA substrates, leading to a hypothesis about how the system may work in vivo. The direct conclusions are well supported by the experimental data. The big unanswered question is whether the results obtained with a minimal Cas1-Cas2 system in vitro reflect what happens in the bacterial cell. More proteins are present in the cell that could modify the reaction and the native DNA array structure differs somewhat from what was used here. Overall, the results presented are intriguing, but their relationship to the cellular mechanism of CRISPR array generation is unclear, since it is not yet known whether DNA elements are inserted in vivo in the same way as observed here in vitro.

Specific comments:

I found the experiment shown in Figure 1F to be difficult to follow from the Figure legend and the text. What reaction(s) were performed for lanes 3-7? The centers of the distributions seem to differ by more than 1 step to me, but I may have misunderstood what is being compared.

Figure 2. What is the evidence that additional product bands differ by 1 and 2 nucleotides for the nicked target? It doesn't look like 3-fold more integration; is this a visual estimate or was quantitation done?

In the case that the second integration step is coupled to disintegration of the first, the authors state that the two integration products would be equal at steady state. The reason for this conclusion should be explained.

Figure 4 – the location of the 4-bp additions to the sequences should be shown.

Figure 8 and discussion. Repair of the product of a second site integration followed by a 3'-OH mediated disintegration at the first site would likely not yield the correct CRISPR array structure. Hydrolysis coupled to an eventual second site integration would yield an intermediate that could be repaired to give the correct array structure. Thus, the reversible disintegration reaction involving the 3'-OH doesn't really seem to play a role in the proposed pathway. The simple conclusion that Cas1/2 catalyzed hydrolysis of a phosphodiester may relieve strain and allow productive transposition to occur doesn't get emphasized enough in my opinion.

---

## [Author Response]

[Editors’ note: The authors appealed the original decision. What follows is the authors’ response to the first round of review.]

Reviewer #1:In this manuscript Ma and colleagues present a biochemical investigation into integration by the Spy Cas1-Cas2 integrase, a key component of the type II CRISPR immune response. From this work they challenge the view that disintegration (the reverse of integration) only functions as an off-pathway reaction (proof reading) and propose that disintegration-promoted integration is functionally important for integration into CRISPR arrays.CRISPR integration proceeds through two steps. In the first, one strand of the prespacer (fragment of foreign DNA) is integrated at the leader-repeat (R-L) junction of the CRISPR array. In the second step, the other strand of the prespacer integrates at the repeat-spacer (R-S) junction resulting in a fully integrated prespacer. The authors propose that, in vitro, water-mediated disintegration at the first site (R-L) is concomitant with second site integration (R-S). The authors present good evidence that this is happening in vitro though a series of elegant biochemical experiments using plasmid and oligo substrates.This is an intriguing hypothesis but its significance is unclear. As the authors themselves acknowledge it could very well be that disintegration is not required in vivo and that this is an in vitro feature of the reaction. Admittedly, demonstrating that disintegration is important in vivo would be a very difficult task. In addition, the in vitro system used here is only partially reconstituted. The substrates lack a PAM sequence, which is necessary for protospacers to be incorporated in the correct orientation and may help direct the first integration event to the L-R junction. Presumably because of this all the reactions presented do not analyze the orientation of the incorporated prespacer sequence. Cas9 and Csn2 are also absent (as are other potentially required host factors), which are necessary for correct integration in vivo.

*Strand specificity*: The in vitro integration reactions with the Cas1-Cas2 complex were done using a protospacer of the optimal size (26 nt on each strand with the four 3’- proximal bases on each strand as unpaired). Either proto-spacer strand is equally competent to initiate the strand transfer reaction, as could be inferred from Figure 3 of the original submission. Here, reactions utilized modified proto-spacers that differed in their top and bottom strand lengths. They gave two insertion products (IP) each at the L-R (leader-repeat) and R-S (repeat-spacer) junctions of a normal target site. In modified targets in which integration was limited to just the L-R junction, two insertion products were formed. One panel of Figure 3 (which is retained in the revised manuscript) showing the four insertion products from the normal target (lane 10) and two from the modified targets (lanes 11-13) for a protospacer with 26 nt and 31 nt long strands.

The ability of either proto-spacer strand to initiate integration is now more directly shown in Figure 2 (new) of the revised manuscript. Here the labeled top or bottom strand of the proto-spacer (PS) gave insertion products (IP) at the L-R and R-S junctions of the target site. Panel B of Figure 2 demonstrates this result.

*Cas9, Csn2 included reactions*: The data for reactions containing Csn2 or Cas9 or both were not shown previously, as they did not alter Cas1-Cas2 activity by promoting strand specificity of integration or suppressing disintegration. These results are now shown in the revised Figure 2 (linear target) and the new Figure 2—figure supplement 1 (supercoiled target).

The relevant revised text describing the lack of strand specificity to proto-spacer integration by Cas1-Cas2 and the Csn2/Cas9 effects on integration is detailed below.

*Page 15, lines 229-235*. “Unlike orientation-specific proto-spacer integration in vivo, Cas1- Cas2 reactions in vitro showed no strand-specificity (Figure 2B). This bias-free insertion of the top or bottom strand from the proto-spacer was unchanged by the addition of Csn2 or Cas9 or both to the reactions (Figure 2C-E). These proteins, singly or in combiantion, also failed to stabilize proto-spacer integrations in the supercoiled plasmid target (Figure 2—figure supplement 1). Instead, they inhibited plasmid relaxation. Inhibition could occur at the level of integration *per se* or strand rotation during integration-disintegration.”

PAM-containing substrates: We have now tested Cas1-Cas2 activity (with and without added Csn2 or Cas9 or both) on PAM-containing substrates that mimic ‘pre-spacers’, Figure 2—figure supplement 3 (new).

In these substrates, a proto-spacer strand of the standard length (26 nt; lacking PAM or its complement) is inserted at the L-R junction with higher efficiency than the longer strand (containing PAM or its complement). Following the first integration at L-R, the pre-spacer mimics containing > 26 nt in one strand or both strands are inhibited in the second strand transfer to the R-S junction.

The revised ‘Results’ section has the following added description of the activities of PAM-containing pre-spacer mimics.

Pages 16-19, lines 265-297. “Cas1-Cas2 activity on pre-spacer mimics carrying the PAM sequence

The strand cleavage and strand transfer steps of proto-spacer insertion at the CRISPR locus must engender safeguards against self-targeting of the inserted spacer as well as its non-functional orientation. […] The mechanism for generating an integration-proficient and orientation-specific proto-spacer, which may not be conserved among CRISPR systems, is poorly understood at this time.”

Figure 1: using a plasmid that lacks a CRISPR array is an important control. It is necessary to establish that the integration events being monitored are indeed at the CRISPR array because formation of half-sites at other locations is expected to result in disintegration. It could also be beneficial to include sequence data to map the site of integration in the pCRISPR targets. This has been done before in this system (PMID: 27595346) but would again bolster the conclusions presented here.

In contrast to the *E. coli* Cas1-Cas2 complex, the S. pyogenes Cas1-Cas2 shows high discrimination between an authentic CRISPR target locus and non-specific DNA for proto-spacer integration (Wright and Doudna; Nat Struct Mol Biol, 2016; 23: 876-883).

In Figure 1—figure supplement 1B, we tested this specificity using two closely related plasmids, pCRISPR (containing the target site) and pUC19 (lacking this site). Plasmid relaxation by Cas1-Cas2, indicating integration-disintegration, is seen only with pCRISPR and not pUC19.

Given the sharply contrasting activities of the target and non-target plasmids in integration-disintegration, additional sequence information is unlikely to shed new insights into this reaction.

Finally, the two product bands of predicted sizes seen in the integration reactions with linear DNA substrates containing the minimal CRISPR locus attest to the selectivity of the Leader- Repeat (L-R) and Repeat-Spacer (R-S) junctions as the targets for proto-spacer integration.

On page 19; the authors state that "…the kinetic correspondence between disintegration and the second integration reactions suggest that the two events are potentially coupled". However, the data presented in Figure 5A do not support this conclusion. The extent of second-site integration (the 81 nt band) and first site disintegration (the 92 nt band) do not change over the measured time frame. Both reactions appear to be over at 1 minute, which is also the first time point. Therefore it is impossible to know if both reactions occur at the same rate or if one is faster than the other.

In the substrate mimicking a semi-integrated proto-spacer at the L-R junction, the disintegration reaction (92 nt product) and the second integration event at the R-S junction (81 nt) are rapid reactions saturating at 1 min (as the reviewer correctly notes). The reverse reintegration from the R-S to the L-R junction is a slower reaction saturating at ~10 min. We agree that product measurements at earlier times than 1 min would be needed to establish strict kinetic correspondence between disintegration and second integration. What we intended to state was the rapid kinetics of R-S integration and disintegration compared to the reverse L-R integration is consistent with the two reactions being coupled. We have now rephrased this sentence as below.

Page 30, lines 431-434. “The prominence of disintegration (DP; 92 nt) and the rapid kinetics of disintegration and the second integration at the R-S junction (IP(R-S); 81 nt) (Figure 5A) suggest that the two events are potentially coupled.”

With the exception of Figure 6, there are no markers, or sequence reaction products, on any of the acrylamide gels presented in the manuscript. It's therefore impossible for the reader to evaluate the size of any of the bands for themselves. This information should be included in any revised manuscript at least at the first instance of a particular sized product.

We have now included size markers in the revised Figure 2 (also Figure 2—figure supplement 3) showing reactions with the linear target site. The markers run in the ‘M’ lanes of these figures are synthetic oligonucleotides of the predicted sequences for proto-spacer strand insertions at the L-R or R-S junction.

We have kept the reaction conditions similar to those used by Wright and Doudna (2016). The migration of the two product bands from the linear target site (Figure 2, for example; see 1B under ‘General comments’, reviewer 1) are consistent only with proto-spacer integration at the L-R junction (IP(L-R)) or at the R-S junction (IP(R-S)). Based on the sizes of the ‘Leader’, ‘Repeat’ and “Spacer’ segments in the target, the sizes of (IP(L-R)) and (IP(R-S)) can only be 89 nt and 81 nt, respectively, as marked in the original Figure 2 (now revised). Furthermore, the authenticity of the integration products is substantiated by the reactions shown in Figure 3. Here, when the target site is modified to alter the spacing between L-R and R-S junctions, the integration products at L-R remain but those at R-S disappear.

The trombone substrates used in Figure 6 are very different from substrates used by Wright and Doudna (2016) or in any other published study. Here size markers were important to verify the prediction that the products with and without disintegration will differ precisely by 26 nt (the size of the pre-integrated proto-spacer strand).

The experiments described in Figure S8 lack a Cas1 dead control. It's important to remove the possibility that the observed cleavage pattern results from a contaminating nuclease.

We did try dCas1-Cas2 in the cleavage reactions using PAM-containing substrates. However, no meaningful conclusion could be drawn because the background of non-specific cleavage (seen with the Cas1-Cas2 complex) was increased in this case. We suspect that dCas1 stimulates endonucleolytic cleavage by Cas2. Note that even in reactions with Cas1-Cas2 (in Figure S8; original number), the PAM-specific cleavage was not exclusive, only significantly above random cleavage.

The cleavage mechanism that generates integration-competent proto-spacer ends is far from settled at present. The *E. coli* Cas1-Cas2 appears to perform this task (Wang et al., 2015, Cell 163: 840-853), as would be consistent with our results shown in Figure S8 using S. pyogenes Cas1-Cas2. However, the latest study attributes cleavage in the S. pyogenes system to Cas9 or to bacterial nuclease(s) that have not been characterized (Jakhanwal, et al., 2021, Nucleic Acids Res, in press). The interpretation of these assays were based on the recovery of integrated proto-spacers in vivo in bacteria transformed with in vitro treated DNA substrates (which further complicates the situation).

Given the uncertainties surrounding pre-spacer to proto-spacer conversion, and the lack of in vitro integration coupled to PAM-specific cleavage, we have removed the original Figure S8 from the revised manuscript. We have now addressed the properties of PAM-containing substrates in the in vitro reactions shown in Figure 2—figure supplement 3 (new) and the corresponding description under ‘Results’ (starting at line 265, page 16; ‘Cas1-Cas2 activity on pre-spacer mimics containing the PAM sequence’).

Some of the experiments are difficult to follow. For example it took me a long time to really understand the experiments in Figure 6, which are really elegant experiments but difficult to follow. I worry that this limits the accessibility of this work. I would encourage the authors to reevaluate how experiments are presented such that they are accessible to a broad audience. It would be helpful to not have to refer to a supplemental figure to understand the main figure i.e. the main figures should be self-contained.

We agree that schematics of the long and short second integration products would make it easier for the reader to follow the experiments and the results. We have now added these schematics in Figure 6B, C against the product bands.

We have also revised and expanded the description of the experimental design to make them more easily accessible to the non-specialist reader. In particular, we have underscored the distinction between integrations of the second proto-spacer strand with or without disintegration of the first. Disintegration shortens the size of the product by the length of the disintegrated strand, 26 nt.

Page 32, lines 478-490. “The predicted products of the second integration event in a trombone substrate would differ by 26 nt (the size of the integrated proto-spacer strand), depending on whether it is associated with disintegration or not. Labeling the 5’-end of the proto-spacer strand integrated at L-R can only reveal integration at R-S without disintegration at L-R. The size of this product analyzed on a denaturing gel will be 183 nt (26 nt + 26 nt protospacer + 36 nt + 36 nt repeat + 10 nt spacer + 30 nt tether + 11 nt leader + 8 nt extension to leader). If integration at R-S were to be accompanied by disintegration at L-R, the predicted 157 nt product would be invisible as the release of the spacer strand (26 nt) results in the loss of the radioactive label as well. Labeling the 3’-end of the bottom strand containing the R-S junction target site (by adding one 32P-labeled nucleotide in a Klenow polymerase reaction) would reveal integration events coupled to disintegration (158 nt) or uncoupled from it (184 nt). Detailed schematic illustrations of the possible products from the 5’ and 3’ end-labelled substrates and their predicted sizes are given in Figure 6—figure supplement 2.”

Reviewer #2:Specific comments:I found the experiment shown in Figure 1F to be difficult to follow from the Figure legend and the text. What reaction(s) were performed for lanes 3-7?

The purpose here was to estimate potential DNA unwinding of the integration target by the bound Cas1-Cas2 complex. Experimentally, the more relaxed topoisomer distribution of the supercoiled target plasmid (containing the CRISPR locus) resulting from the proto-spacer plus Cas1-Cas2 reaction is compared to the equilibrium relaxed distribution formed by strand sealing of the nicked form of the same plasmid by T4 ligase. The difference between the two distributions gives the torsional stress (writhe plus twist) sequestered by the interaction of Cas1-Cas2 with its target. Lane 3 of Figure 1F shows the reference relaxed distribution formed upon nick closure by ligase. Lanes 4-7 show the distributions formed when the nicked plasmid pre-incubated with the indicated proteins (with or without the proto-spacer) is sealed by ligase. The center of the topoisomer distributions in lanes 3-7 is very nearly the same, indicating that any topological constraint imposed by Cas1-Cas2 on the target is dissipated via the nick before strand sealing. Following incubation with Cas1-Cas2/proto-spacer, the supercoiled plasmid is relaxed (lane 8), but not to the same extent as the nicked plasmid sealed by ligase. The distribution formed by Cas1-Cas2 is more negatively supercoiled, indicating DNA unwinding by Cas1-Cas2.

We have now explained the experimental rationale in more detail as follows.

Page 7, lines 163-169. “We probed linking number changes (ΔLks) accompanying Cas1- Cas2-mediated integration-disintegration in a supercoiled target plasmid and T4 ligase-mediated strand closure in the nicked form of the same plasmid with or without bound Cas1-Cas2. Provided the Cas1-Cas2-induced topological stress within a target site is stable throughout the reaction, it will be preserved in the plasmid topoisomers formed by integration-disintegration (Figure 1B, C). In principle, the target topology in a Cas1-Cas2-bound nicked plasmid may also be captured by resealing the nick with DNA ligase.”

Page 11, lines 194-204. “Strand sealing in the nicked plasmid was performed by T4 ligase without added factors (Figure 1F; lane 3) or after pre-incubation with the indicated Cas proteins and/or the proto-spacer (Figure 1F; lanes 4-7). Nearly identical topoisomer distributions were obtained after ligation in the absence of Cas1-Cas2 (Figure 1F; lane 3) or in the presence of either Cas1-Cas2 (Figure 1F; lanes 5) or the catalytically inactive dCas1-Cas2 (Figure 1F; lane 6).

Addition of protospacer by itself (Figure 1F; lane 4) or together with dCas1-Cas2 (Figure 1F; lane 7) to the ligase reaction did not alter this distribution. The topoisomer distribution of the Cas1-Cas2 plus proto-spacer treated supercoiled plasmid (Figure 1F; lane 8) was more negatively supercoiled (ΔLk between -1 and -2) than that obtained from nick closure in the presence of dCas1-Cas2 and proto-spacer (Figure 1F; lane 7) (Figure 1—figure supplement 2).”

The centers of the distributions seem to differ by more than 1 step to me, but I may have misunderstood what is being compared.

Marking the centers of the topoisomer distributions is tricky because of the inherent difficulty in adjusting the base lines for the peaks in the densitometric tracings. The individual topoisomers in the Cas1-Cas2 reaction are offset by ΔLk = ~0.5 relative to those in the ligase reaction (note the stagger between the bands in lanes 7 and 8 in Figure 1F). The difference in the relaxed distributions formed by ligase and by Cas1-Cas2 is between 1 and 2 negative supercoils. This ΔLk value is indicated in Figure S2 , and noted in the revised text detailed below:

Page 11, lines 201-204. “The topoisomer distribution of the Cas1-Cas2 plus proto-spacer treated supercoiled plasmid (Figure 1F; lane 8) was more negatively supercoiled (ΔLk between -1 and -2) than that obtained from nick closure in the presence of dCas1-Cas2 and proto-spacer (Figure 1F; lane 7) (Figure 1—figure supplement 2).”

Pages 11-12, lines 209-214. “The observed ΔLk between the Cas1-Cas2 relaxed and T4 ligase sealed plasmid distributions (Figure 1F; Figure 1—figure supplement 2) is consistent with the unwinding of a little more than one DNA turn within the target sequence during proto-spacer integration. This torsional strain may promote the formation of the repair-ready intermediate in which single stranded repeat sequences flank the inserted proto-spacer (Figure 1A).”

Figure 2. What is the evidence that additional product bands differ by 1 and 2 nucleotides for the nicked target?

We have not characterized the additional product bands in the reactions with the nicked target site (now Figure 2—figure supplement 2). However, a reasonable explanation for their formation, consistent with their gel mobilities, is strand transfer to either of the two phosphodiester bonds between the authentic phosphodiester target and the nick.

In the schematic diagram of the nicked target (from Figure S4), the normal target phosphodiester (p) and the two neighboring ones (p) towards the nick (marked by the short vertical arrow) are indicated on the top and bottom strands. The sequences flanking the nicks are shown in red for the leader (L), yellow for the repeat (R) and blue for the spacer (S). The normal proto-spacer integrant at the L-R junction and the aberrant ones (one nucleotide or two nucleotides into the leader) are 89, 90 and 91 nt long, respectively. The corresponding lengths for the integrants at the R-S junction and at the adjacent phosphodiesters in the spacer are 81, 82 and 83 nt.

The potential origin of the extra integration products from the nicked target is explained under “Results’ and in Supplementary Text S2 associated with the legend to Figure 2—figure supplement 2.

Pages 15-16, lines 243-249. “The nick-containing target yielded, in addition to the 81 nt and 89 nt products expected from the standard linear target (Figure 2—figure supplement 2A, B), product bands migrating slightly higher than each (Figure 2—figure supplement 2A, C). These extra bands, though uncharacterized, are consistent with potential integration events at the two phosphodiester positions between an authentic phosphodiester target and its proximal nick (Supplementary Text S2 under Figure 2—figure supplement 2). The flexibility afforded by the nick could presumably facilitate these aberrant integration events.

Pages 53, lines 889-895. “Text S2. The uncharacterized doublet bands migrating above the 81nt and 89 nt products could indicate integrations at phosphodiester positions marked as ‘p’ in the nicked target. The expected product lengths for the upper doublet are 90 nt and 91 nt, and those for the lower doublet are 82 nt and 83 nt. The slightly uneven spacing of the 82 nt and 83 nt bands from the 81 nt band is likely due to the size difference in the additional base(s) they harbor. The 82 nt and 81 nt products differ by an extra ‘A’ gained from the spacer. The second additional base present in the 83 nt product is the smaller ‘C’.”

It doesn't look like 3-fold more integration; is this a visual estimate or was quantitation done?

The quantitation for the nicked target includes the authentic plus aberrant integration products; that for the unnicked target comprises only the authentic products (no aberrant products are formed in this reaction). The estimated difference in total product yields was between 2- and 3-fold (closer to 3). We have now changed ‘~3 fold’ to ‘2- to 3-fold’:

Page 16, lines 249-251. “The amount of the integration product (normal plus aberrant) was increased 2- to 3-fold in comparison to the normal product formed from the target without a nick.”

In the case that the second integration step is coupled to disintegration of the first, the authors state that the two integration products would be equal at steady state. The reason for this conclusion should be explained.

The initial integration of the proto-spacer follows the order, L-R first and R-S next. However, as the present work demonstrates, integration at R-S is most often associated with disintegration at L-R (mediated by the 3’-OH or by water). Once integrated at R-S, the protospacer strand can reverse-integrate at L-R with accompanying disintegration at R-S. Over time, these oscillations will equalize integrations at L-R and R-S, even though at early times integrations at L-R would exceed those at R-S.

We have now added the following clarification to the text.

Page 16, lines 256-260. “A third possibility is that, within an intermediate containing a semi-integrant, integration of the second strand is coupled to disintegration of the first. In this case, the one-ended integrant will oscillate between the L-R and R-S junctions. As a result, the net integration events at L-R and R-S will be equal at steady state.”

Figure 4 – the location of the 4-bp additions to the sequences should be shown.

We have now provided the repeat sequences present in the different L-R and R-S half-target sites. To avoid overcrowding of the main figure, this information is included as a new supplementary figure (Figure 4—figure supplement 1).

Figure 8 and discussion. Repair of the product of a second site integration followed by a 3'-OH mediated disintegration at the first site would likely not yield the correct CRISPR array structure. Hydrolysis coupled to an eventual second site integration would yield an intermediate that could be repaired to give the correct array structure. Thus, the reversible disintegration reaction involving the 3'-OH doesn't really seem to play a role in the proposed pathway. The simple conclusion that Cas1/2 catalyzed hydrolysis of a phosphodiester may relieve strain and allow productive transposition to occur doesn't get emphasized enough in my opinion.

The reviewer correctly notes that hydrolysis coupled to the second integration (or disintegration by water nucleophile) is crucial in generating the intermediate that can be repaired to give the correct proto-spacer insertion without disarranging the CRISPR array. However, we disagree with the assertion that disintegration carried out by the 3’-OH has no role in proto-spacer insertion. To simplify the explanation, we have split the relevant parts of Figure 8 into panels A-C.

Panel A shows disintegration by water at the leader-repeat (L-R) junction—the first integration site--and integration at the repeat-spacer (R-S) junction (the second integration site).

This generates the proper DNA intermediate for repair (as the reviewer concurs). Panel B shows the same second integration reaction, this time disintegration at L-R being mediated by the 3’-OH. This step by itself cannot promote proto-spacer integration by DNA repair (as the reviewer rightly argues). However, the reaction restores the L-R junction, making integration in the R-S to L-R direction now possible (panel C). This reverse integration, accompanied by water mediated disintegration at R-S, will generate a productive repair intermediate (analogous to that in A). Thus, both types of disintegration (hydrolysis and transesterification) contribute to proto-spacer insertion, though in quite different ways.

The relevant text under ‘Discussion’ and revisions in the legend to Figure 8 further clarify this explanation.

Page 41, lines 580-590. “Water-mediated disintegration of the proto-spacer at the L-R junction during strand transfer to the R-S junction will produce an intermediate with a gap in the bottom strand and a gap plus nick in the top strand (Figure 8; left). An analogous intermediate with the gap plus nick in the bottom strand may also be formed (Figure 8; right). Here, a proto-spacer is first transferred to the R-S junction with concomitant 3’-hydroxyl-mediated disintegration from the L-R junction (and restoration of this junction). This is followed by a subsequent integration in the reverse direction, with disintegration at the R-S junction promoted by water. The spacer insertion event can be completed by gap-filling/displacement DNA synthesis followed by ligation (Figure 8). Thus, disintegration reactions mediated by the 3’-hydroxyl and by water are both relevant, though in mechanistically distinct ways, to the rescue of semi-integrated proto-spacers into fully integrated spacers.”

Page 47, lines 741-751. “Figure 8 legend. A model for disintegration-assisted integration. Integration of the proto-spacer by Cas1-Cas2 is predominantly one-ended. Following the initial strand transfer to the L-R junction, transfer of the second strand to the R-S junction is associated with disintegration at L-R. Oscillations of the semi-integrated proto-spacer become possible when disintegration mediated by the 3’-hydroxyl reforms the L-R or R-S junction at the disintegration site. Disintegration mediated by water during a second strand transfer step will block oscillations. However, the nick between L and R or R and S will be preserved, and can be utilized by DNA repair machineries to complete proto-spacer insertion. Formation of the repair-suitable intermediate shown at the left is dependent only on water-mediated disintegration, the 3’-hydroxyl-mediated disintegration being dispensable. However, both types of disintegration are required to generate the intermediate shown at the right.”